# Evaluating China's anthropogenic $CO_2$ emissions inventories: a northern China case-study using continuous surface observations from 2005-2009.

Archana Dayalu[1,*], J. William Munger[2,3], Yuxuan Wang[4,5], Steven C. Wofsy[2,3], Yu Zhao[6], Thomas Nehrkorn[1], Chris Nielsen[3], Michael B. McElroy[3], Rachel Chang[7]

[1]Atmospheric and Environmental Research, Lexington, MA, USA
[2]Earth and Planetary Sciences, Harvard University, Cambridge, MA, USA
[3]School of Engineering and Applied Sciences, Harvard University, Cambridge, MA, USA
[4]Department of Earth and Atmospheric Sciences, University of Houston, Houston, TX, USA
[5]Department of Earth System Sciences, Tsinghua University, Beijing, China
[6]School of the Environment, Nanjing University, Nanjing, China
[7]Department of Physics and Atmospheric Science, Dalhousie University, Halifax, Canada
*Formerly at Earth and Planetary Sciences, Harvard University, Cambridge, MA, USA

*Correspondence to*: Archana Dayalu (adayalu@aer.com)

**Abstract.** China has pledged reduction of carbon dioxide ($CO_2$) emissions per unit GDP by 60-65% relative to 2005 levels, and to peak carbon emissions overall by 2030. However, the lack of observational data and disagreement among the many available inventories makes it difficult for China
to track progress toward these goals and evaluate the efficacy of control measures. To demonstrate the value of atmospheric observations for constraining $CO_2$ inventories we track the ability of $CO_2$ concentrations predicted from three different $CO_2$ inventories to match a unique multi-year continuous record of atmospheric $CO_2$. Our analysis time window includes the key commitment period for the Paris accords (2005) and the Beijing Olympics (2008). One inventory is China-specific and two are spatial
subsets of global inventories. The inventories differ in spatial resolution, basis in national or subnational statistics, and reliance on global or China-specific emission factors. We use a unique set of historical atmospheric observations from 2005–2009 to evaluate the three $CO_2$ emissions inventories within China's heavily industrialized and populated Northern region accounting for ~33–41 % of national emissions. Each anthropogenic inventory is combined with estimates of biogenic $CO_2$ within a high-
resolution atmospheric transport framework to model the time series of $CO_2$ observations. To convert the model-observation mismatch from mixing ratio to mass emission rates we distribute it over a region encompassing 90% of the total surface influence in seasonal (annual) averaged back-trajectory footprints (L_0.90 region). The L_0.90 region roughly corresponds to northern China. Except for the peak growing season, where assessment of anthropogenic emissions is entangled with the strong
vegetation signal, we find the China-specific inventory based on subnational data and domestic field-

studies agrees significantly better with observations than the global inventories at all timescales. Averaged over the study time period, the unscaled China-specific inventory reports substantially larger annual emissions for northern China (30%) and China as a whole (20%) than the two unscaled global inventories. Our results, exploiting a robust timeseries of continuous observations, lend support to the

rates and geographic distribution in the China-specific inventory Though even long-term observations at a single site reveal differences among inventories, exploring inventory discrepancy over all of China requires a denser observational network in future efforts to measure and verify $CO_2$ emissions for China both regionally and nationally. We find that carbon intensity in the northern China region has decreased by 47% from 2005 to 2009, from approximately $4kgCO_2/USD_{PPP}$ in 2005 to about $2kgCO_2/USD_{PPP}$ in

2009 (Figure 9c). However, the corresponding 18% increase in absolute emissions over the same time period affirms a critical point that carbon intensity targets in emerging economies can be at odds with making real climate progress. Our results provide an important quantification of model-observation mismatch, supporting the increased use and development of China-specific inventories in tracking China's progress as a whole towards reducing emissions. We emphasize that this work presents a

methodology for extending the analysis to other inventories and is intended to be a comparison of a subset of anthropogenic $CO_2$ emissions rates from inventories that were readily available at the time this research began. For this study's analysis time period, there was not enough spatially distinct observational data to conduct an optimization of the inventories. The primary intent of the comparisons presented here is not to judge specific inventories, but to demonstrate that even a single site with a long

record of high time resolution observations can identify major differences among inventories that manifest as biases in the model-data comparison. This study provides a baseline analysis for evaluating emissions from a small but important region within China, as well a guide for determining optimal locations for future ground-based measurement sites.

# 1 Introduction

China's contribution to world $CO_2$ emissions has been steadily growing, becoming the largest in the world in 2006. China has accounted for 60% of the overall growth in global $CO_2$ emissions over the past 15 years (EIA, 2017). Under the United Nations Framework Convention on Climate Change (UNFCCC) 2015 Paris Climate Agreement, China has committed to reduce its carbon intensity ($CO_2$ emissions per unit GDP) by 60-65% relative to the baseline year of 2005, and to peak carbon emissions

overall by or before 2030. Demonstration of progress on emissions reduction and evaluation of how well specific policies are working is hindered by large uncertainty in the existing Chinese emission inventories. In 2012 the discrepancy between data reported at national and provincial levels was approximately half of China's 2020 emission reduction goals (EIA, 2017; NDRC, 2015; Guan et al., 2012; Zhao et al., 2012). Moreover, China is under mounting pressure to address severe regional air

pollution events that are often associated with $CO_2$ emissions sources—vehicles, power plants and other fossil fuel-burning operations. China's 11[th] Five Year Plan (11[th] FYP) of 2006-2010 included aggressive measures to retire inefficient coal-fired power plants and improve energy efficiency in other industries starting in 2007 (Zhao et al., 2013; Nielsen & Ho, 2013). A number of pollution control measures that were implemented specifically in preparation for the 2008 Beijing Summer Olympics were also largely

in effect by the end of 2007 (Nielsen & Ho, 2013; Wang et al., 2010).

A variety of top-down approaches including inverse analysis (Le Quere et al., 2016) and comparison between atmospheric observations and Eulerian forward model predictions (Wang, X. et al., 2013) have been used to evaluate and constrain emission estimates, albeit at coarse spatial resolution. As noted by

Wang et al. (2011) grid-based atmospheric models have difficulty in simulating high-concentration pollution plumes at specific receptor sites that are too near the source region. The expanding network of high accuracy $CO_2$ observations coupled with high spatial resolution transport models is emerging as a viable tool for evaluating high resolution emission inventories (e.g. Sargent et al., 2018). In this paper we adopt a Lagrangian transport model to simulate atmospheric mixing and transport. Continuous

observations of $CO_2$ for the period 2005-2009 at Miyun, an atmospheric observatory about 100km NE of Beijing provide a top-down constraint for evaluating persistent bias among emissions rates obtained from a suite of three independent anthropogenic emission inventories that were readily available as spatially gridded fluxes.

The three inventories that are evaluated span a range of bottom-up inventory approaches. They are not intended to be an exhaustive set, but are examples to demonstrate the capability to identify significant differences in the ability of different inventories to match the long time series of observations. Emerging inventory approaches based on updated (yet non-China-specific) point-source data and satellite-observations of night lights as a proxy for spatial allocation of energy production (Oda et al., 2018)

were not readily available when this analysis began. Two of the inventories, the Emissions Database for Global Atmospheric Research (EDGAR; European Commission, 2013) and Carbon Dioxide Information Analysis Center (CDIAC), are spatial subsets from larger global models of $CO_2$ emissions

(PBL, 2013; Andres et al., 2016). They rely on national-level energy statistics and global default values for sectoral emission factors, and they estimate activity levels using generalized proxies (e.g. population). The third inventory (ZHAO) is specific to China, with greater reliance on energy statistics at provincial and individual facility levels as well as emission factors from domestic field studies (Zhao et al., 2012). The ZHAO inventory was readily accessible at the time of this research and represents increased efforts in recent years to incorporate more China-specific data into emissions inventories. Other China-specific inventories that have been recently developed but were not readily available at the time of this research include the Multi-resolution Emissions Inventory (MEIC, http://www.meicmodel.org/) and an inventory by Shan et al., 2016. The primary intent of the comparisons presented here is not to judge specific inventories, but to demonstrate that even a single site with a long record of high time resolution observations can identify the potential impact of major differences among inventories that manifest as biases in the model-data comparison.

A study by Turnbull et al. (2011) used weekly flask observations to evaluate a hybrid approach to inventory construction where CDIAC and EDGAR estimates were spatially allocated to a provincial emissions-based grid. However, to our knowledge, none of the truly China-specific $CO_2$ inventories have been evaluated with independent high-temporal resolution atmospheric observations. The official national total for China's 2005 $CO_2$ emissions from energy related activities, used as the benchmark for the Paris commitment, is approximately 5.4Gton $CO_2$ (NDRC, 2015). ZHAO, EDGAR, and the CDIAC national total (Boden et al., 2016) report total 2005 energy-related $CO_2$ emissions that are higher by 31% (7.1Gton), 9%(5.9Gton), and 7%(5.8Gton), respectively. As the official national total is not available in a spatially allocated format, it cannot be tested by observations and we refer to it only as a benchmark in our analysis. We will show that the China-specific inventory (ZHAO) provides excellent agreement with observations, and markedly more so than EDGAR and CDIAC. The result provides guidance for efforts to assess China's emissions at larger scales as well as potential updates for the Paris agreement base year emissions.

In order to independently evaluate and scale existing bottom-up estimates of China's $CO_2$ emissions, we employ a top-down approach using five years of continuous $CO_2$ observations. Modeled concentrations of $CO_2$ are obtained from convolving hourly $CO_2$ surface flux estimates with surface influence estimates ("footprints") derived from the Stochastic Time-Inverted Lagrangian Transport Model driven with meteorology from the Weather Research and Forecasting Model version 3.6.1 (WRF-STILT; Lin et al., 2003; Nehrkorn et al., 2010). NOAA CarbonTracker (CT2015) provides modeled estimates of advected upwind background concentrations of $CO_2$ that are enhanced or depleted by processes in the study region. As atmospheric $CO_2$ concentrations are significantly modulated by photosynthetic and respiratory fluxes, we additionally prescribe hourly biosphere fluxes of $CO_2$ using data-driven outputs from the Vegetation, Photosynthesis, and Respiration Model (VPRM) adapted for China (Mahadevan et al., 2012; Dayalu et al., 2018). VPRM provides a functional representation of biosphere fluxes based on data from remote sensing platforms and eddy flux towers, with significantly better observationally-validated performance relative to subsets of global vegetation models (Dayalu et al., 2018). The WRF-

STILT-VPRM framework has been successfully adapted for similar emissions evaluation studies in North America in regions where biogenic fluxes dominate surface processes (e.g., Sargent et al., 2018; Karion et al. 2016; Matross et al., 2008). For the Northern China region, anthropogenic fluxes exceed biogenic fluxes for all but the peak of growing season, when they are roughly comparable (Dayalu et al., 2018), which reduces the magnitude of overall error from incorrect modeling of the biosphere. In contrast to extensive measurement networks that exist in North America, continuous high-temporal resolution measurements of $CO_2$ necessary for inventory evaluation applications are sparse and very few datasets are available in China (Wang et al. 2010). Despite this limitation, our site provides valuable information and constraints on emissions inventories: the long time series and spatial sampling heterogeneities where the site receives both clean continental air as well as air from one of the heaviest emitting regions of China, present a powerful and unique dataset for the region. Our inventory scaling is confined to the Northern China region, but this region accounts for 33-41% of China's total annual $CO_2$ emissions from fossil-fuel combustion. Model-observation mismatches can be converted from concentration units (ppm) to mass units (Mton $CO_2$) across the most relevant area subset from modeled annual average surface sensitivity footprints ($\mu mol^{-1} m^2 s$). Ultimately, we compare the inventories by quantifying model-observation mismatch for seasons (using additive mass units) and annually (using scaling factors). We note that identical transport fields and modeled biogenic fluxes are applied to all the anthropogenic emission fields. Unresolved transport error and error in biogenic fluxes undoubtedly contributes to scatter in the model-data comparison. While random transport errors are unlikely to generate consistent biases among the inventories, systematic transport errors can be attributed to biases among inventories with differing spatial allocations. Although the interaction of systematic transport errors with differences in spatial distribution could bias individual observations, averaging over longer timescales (seasons, years) minimizes the bias of individual points. With the available observational data it is not possible to evaluate the error in spatial allocation of individual emissions inventories. For example, future access to total column measurements and/or aircraft vertical profiles would provide additional constraints on spatial allocations of sources and sinks.

Section 2 of this paper describes the observational $CO_2$ record used in this analysis. Section 3 details the analysis methods, including WRF-STILT model configuration, a discussion of the main features of the inventories, error evaluation, and inventory scaling methods. We present the results in Sect. 4, beginning with an assessment of seasonality impacts. We then compare inventory performance against observations across multiple timescales from hourly to annual. We conclude Sect. 4 with scaling results, and a brief examination of regional carbon intensity over the study period. Concluding remarks are provided in Sect. 5. Additional methodological details are provided in the accompanying Supplementary Information (SI) and at https://doi.org/10.7910/DVN/OJESO0.

## 2 CO$_2$ observations

This study uses five years (2005-2009) of continuous hourly averaged CO$_2$ observations (LI-COR Biosciences Li-7000; 2-$\sigma$ analytical precision of 0.08ppm), measured at a site in Northern China (Miyun; 40°29'N, 116°46.45'E). The Miyun receptor is an atmospheric measurement station in a rural site 100 km northeast of the Beijing urban center (Fig. SI S2). It was established in 2004 by collaborating researchers at the Harvard China Project and operated by researchers at Tsinghua University. The site is strategically located to capture both clean continental background air from the west/northwest and polluted air from the Beijing region to the southwest. Miyun is located south of the foothills of the Yan mountains; the region consists of grasslands, small-scale agriculture intermingled with rural villages and manufacturing complexes, and mixed temperate forest. Land use grades from rural to suburban and dense urban to the south towards Beijing center and sparsely populated and wooded mountains to the north and west. Further descriptions of the site and details of the instrumentation including calibration strategy and assessment of long-term drifts  are in provided in Wang et al. (2010). Average annual data coverage (based on hourly data) over the study time period was 83% (range: 78% to 92%).

## 3 Methods

We evaluate the performance of the ZHAO, EDGAR, and CDIAC inventories coupled with biogenic fluxes by modelling five years of hourly CO$_2$ observations using the Stochastic Time-Inverted Lagrangian Transport Model (STILT; Lin et al., 2003) run in backward time mode driven by high resolution meteorology from the Weather Research and Forecasting Model version 3.6.1 (WRF). The WRF-STILT tool models the surfaces that influenced each measurement hour in the study domain (Figure 1). Hourly vegetation CO$_2$ fluxes are prescribed by the VPRM adapted for China (Mahadevan et al., 2008, Dayalu et al., 2018). We categorize seasons by months based on regional growing season patterns, which are heavily dominated by winter wheat/corn dual-cropping regions in the North China Plain (Dayalu et al. 2018). Winter wheat emergence in the spring and corn emergence in later summer shift the seasonal patterns such that regional seasons are more appropriately represented as January, February, March (JFM/Winter); April, May, June (AMJ/Spring); July, August, September (JAS/Summer); and October, November, December (OND/Fall).

Ultimately, modeled concentrations of CO$_2$ are obtained from convolving hourly surface flux estimates with footprints derived from the WRF-STILT framework. NOAA CarbonTracker (CT2015) provides estimates of advected upwind background concentrations of CO$_2$ that are enhanced or depleted by processes in the study region. Our final modeled-measurement data set is the subset consisting of local daytime values (hourly data from 1100h to 1600h). Of this subset, only individual hours for which observational data exists (i.e., non-missing data) is included. The final data set was further filtered to include only CT2015 background values satisfying true background criteria as described in Sect. 3.5

and in the SI, Sect. S4. As is typical for studies of this nature, our analysis focuses on observations
during the 1100 to 1600 local time period. The stronger vertical mixing in the daytime atmosphere
(notably absent at night) reduces the influence of extremely local emissions. We select the 1100-1600
window to avoid the presence of shallow inversion layers that are poorly represented in STILT and use
the period when vertical mixing through the entire boundary layer is at its maximum (McKain et al.,
2015; Sargent et al., 2018). We adjust fluxes based on model-measurement mismatch of this final data
subset, focusing on the region that we model as most influential to the signal measured at the receptor.
Method details and model components are described individually below.

## 3.1 WRF-STILT Model Configuration

The WRF-STILT particle transport framework and optimal configuration have been extensively tested
in several studies using mid-latitude receptors (e.g., Sargent et al., 2018; McKain et al., 2014; Kort et

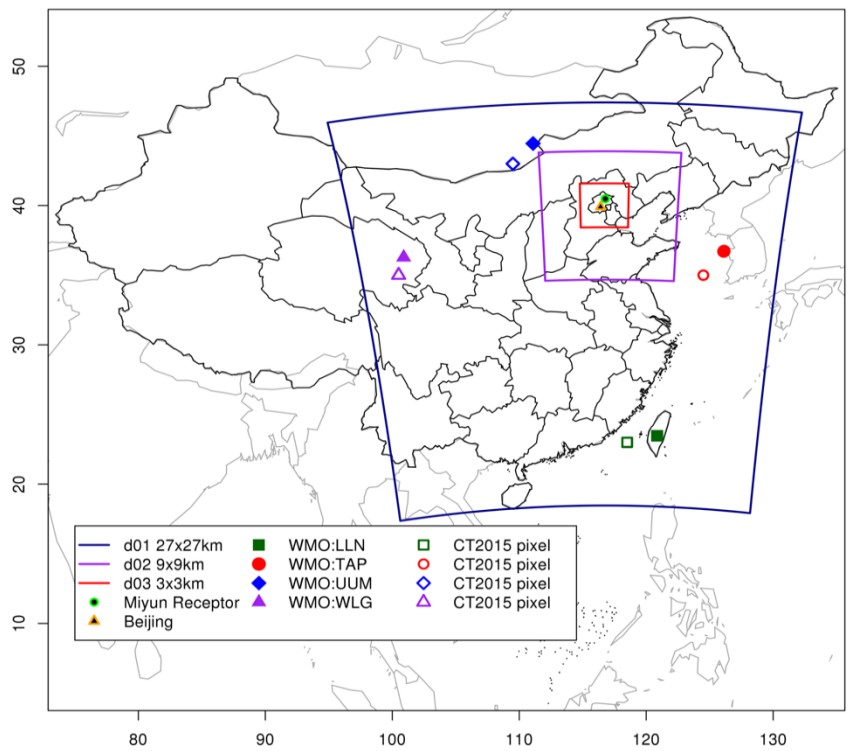

**Figure 1.** Study domain configuration. Miyun receptor and Beijing center are
located within the innermost domain at a resolution of 3x3km. NOAA
ESRL/WMO (WMO) flask sampling sites used to evaluate bias in CT2015
modeled backgrounds are the solid shapes; nearest CT2015 comparison pixel is the
corresponding unfilled shape.

al., 2013; McKain et al. 2012; Miller et al., 2012). WRF is configured with 41 vertical levels and two-way nesting in three domains, with the outermost domain covering nearly seven administrative regions (Figure 1, Figure 2), defined according to convention in Piao et al. (2009). The domain resolutions from coarsest to finest are 27km (d01), 9km (d02), and 3km (d03). Initial and lateral WRF boundary conditions are provided by NCEP FNL Operational Model Global Tropospheric Analyses at 1°x1°

spatial 6-hourly temporal resolution (NCEP, 1999). Nudging of fields is implemented in the outer domain only, and never within the Planetary Boundary Layer (PBL). WRF output is evaluated against

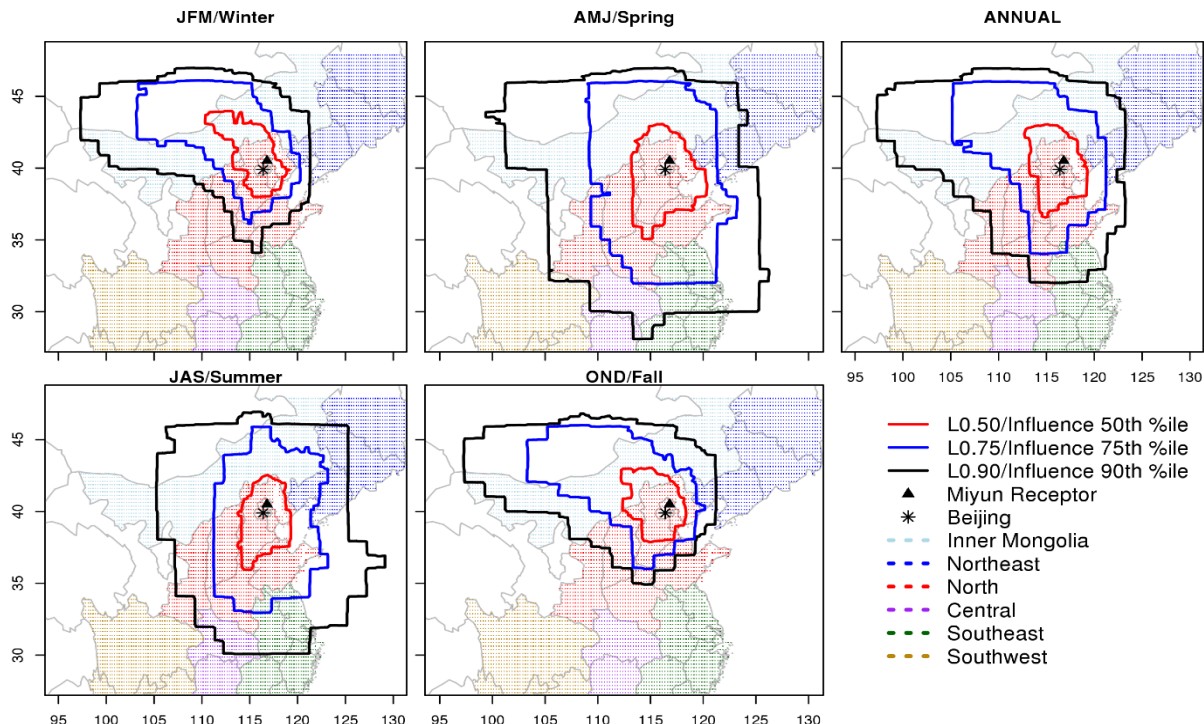

**Figure 2.** 2005-2009 mean seasonal (a-d) and Annual (e) footprint contours, as percentiles of influence highlighted by administrative region. Red, blue, and black contour lines represent 50th, 75th, and 90th percentile regions, respectively. Stippling represents location of 0.25º x 0.25º footprint and inventory gridcell centers, colored by relevant administrative regions. Northern China (red stippling) is the administrative region with predominant influence on Miyun observations, followed by Inner Mongolia and Northeast China. Southeast and Central China have minimal representation, and only during the spring and summer seasons.

publicly accessible 24-hourly averaged observational datasets from the Chinese Meteorological Administration (CMA); finer temporal resolution meteorological data is not publicly available. WRF run details are presented in Dayalu (2017) and at http://dx.doi.org/10.7910/DVN/OJESO0. A snapshot

of results from comparison with China Meteorological Administration ground-station measurements is presented in SI Sect. S1 and Figures S1-S4.

The STILT model is configured in backward time mode. The particle release point is set as the Miyun measurement sample inlet (the receptor). The inlet height is 158m above sea level (masl), corresponding
to 6m above ground level (magl). In our study, the hilltop site was located in an area where the surrounding land was not very productive or intensively cultivated (SI Fig. S2). There is a long history of using short towers in low productivity areas for regional studies (e.g. NOAA Earth Systems Research Laboratory—NOAA ESRL Barrow, Alaska observatory at 11 magl). In addition, the station is located on a small hilltop, so even though the actual inlet height above ground is low, it has a topographic
advantage in that it effectively samples air from a greater height relative to the surroundings. Topographic advantage was exploited in a similar manner in Karion et al. (2016) in the context of an Alaskan $CO_2$ study. However, Karion et al. (2016) were able to use a suite of additional data to confirm the validity of their assumption including comparisons to concurrent aircraft measurements and multiple inlets at 31.7magl, 17.1magl, and 4.9magl. In our study, independent verification from concurrent
aircraft measurements (for example) or multi-level inlet locations were not available to quantify the impact of absolute and relative inlet location on transport uncertainty.

Each hourly footprint ($CO_2$ concentration attributed to each unit of flux as ppm $\mu mol^{-1}\, m^2\, s$) provides an estimate of surface influence on the measurement and is calculated from releasing 500 particles from
the measurement site (receptor) until they reach the outer domain boundaries up to seven days back in time. The STILT 0.25º x 0.25º footprint map for each measurement hour up to 7 days back in time enables assessment of regions in the study domain to which the receptor is most sensitive. These entire gridded footprints are convolved with anthropogenic and biogenic $CO_2$ flux estimates to provide a final modeled concentration (ppm) of $CO_2$ at the receptor. For clarity, we display the regions of importance to
the receptor based on contours calculated from the overall STILT footprints at the 50th (L_0.50 region), 75th (L_0.75 region), and 90th (L_0.90 region) percentile levels (Figure 2). The percentile contours are calculated as follows: the average (seasonal, annual) footprints from 2005 to 2009 are ordered from high to low. We multiply each fraction (0.5,0.75,0.9) with the summed footprints and use cumulative sums of the ordered footprints as a guide to select all points with influence magnitude equal to or
greater than this cutoff value. SI Figure 11 illustrates a single footprint map along with the average influence and a plot of cumulative influence to demonstrate the percentile level selection process. We emphasize that we use the entire STILT footprint convolved with fluxes to estimate the receptor $CO_2$ concentration. We only use the L_0.90 region to provide a reasonable area across which to ascribe the effective inventory adjustment (converted from ppm model-observation mismatch to mass units). As SI
Figure 11c shows, the L_0.90 region strikes a balance between capturing sufficient influence while avoiding an unrealistically large adjustment region for a single observation site. Conversely, corrections based on the smaller L_0.75 region would include larger uncertainties from the diffuse influence of emissions outside the L_0.75 region (not accounting for 25% of average surface sensitivity), yet the model-observation mismatch would be ascribed to a region approximately half the area of the L_0.90

region. Deriving correction factors based on integration over the entire L_0.90 region is a more conservative approach where the model-observation mismatch in mass units is distributed over a larger area.

Further model details are available in SI Sect. S2. Complete WRF-STILT settings and STILT footprint
files are available from http://dx.doi.org/10.7910/DVN/OJESO0.

### 3.3 Anthropogenic $CO_2$ Emissions Inventories

ZHAO, EDGAR, and CDIAC report estimates of total annual emissions of $CO_2$ at 0.25º x 0.25º, 0.1º x 0.1º, and 1º x 1º original grid resolutions, respectively. We regridded the EDGAR and CDIAC
inventories to the 0.25º x 0.25º resolution, using NCAR Command Language version 6.2.1 Earth System Modeling Framework conserve regridding algorithm to preserve the integral of emissions (Brown et al., 2012). Differences between annual total emissions for EDGAR and CDIAC inventories introduced by regridding are smaller than the interannual trends or differences between the inventories (SI Sect. S3 and Figure S5). We present the main components and defining features of the three
anthropogenic $CO_2$ inventories below.

The ZHAO inventory provides estimates of total annual emissions for 2005 through 2009. In addition, spatial location of emissions is given for years 2005 and 2009 on a 0.25º x 0.25º grid. Using 2005 and 2009 gridded values, we calculate an average percent contribution of each grid cell to the total
emissions. The average contributions are used as weights to spatially allocate 2006, 2007, and 2008 total annual emissions. We evaluate and justify this assumption in detail in SI Sect. S3 and Figure S6. The ZHAO inventory represents one of the first statistically rigorous bottom-up $CO_2$ inventories for China. It relies on provincial- and facility-level data rather than national level data, which has been noted previously as major uncertainty in Chinese emission inventories; total $CO_2$ emissions estimates
based on provincial data are typically higher than those using national statistics (Zhao et al., 2013). Satellite observations of criteria air pollutants (e.g., nitrogen dioxide, which serves as a proxy for fossil fuel combustion) show greater agreement with provincial statistics (Zhao et al., 2012). The increased use of China-specific emission factors and activity levels based on domestic field studies is a shift from other inventories that rely heavily on global averages to estimate processes occurring in China. Despite
the increased incorporation of China-specific field data, the largest sources of uncertainty to the ZHAO inventory are industrial emission factors, and activity levels across all sectors. Total uncertainty in the inventory is estimated as -9% to +11%. (Zhao et al., 2012).

The EDGAR emissions database continues to be a major prior in atmospheric studies, and the $CO_2$
inventory is used to inform key global scientific results considered by the UNFCCC Conference of Parties. The EDGAR global inventory (atemporal EDGAR v4.2 FT2010 gridded emissions) takes total annual estimates of national emissions and downscales emissions to a 0.1º x 0.1º as a function of

road/shipping networks, population density, energy/manufacturing point sources, and agricultural land. Estimates for China are available for all five years as gridded inventories. Reported uncertainties for global emissions are ±10% (http://themasites.pbl.nl/tridion/en/themasites/edgar/documentation/uncertainties/index-2.html). However, this applies to global averaged uncertainty; we expect uncertainty for China to be much higher.

We include the CDIAC inventory here due to its historical prevalence as a benchmark inventory for global indicators, including evaluations of carbon intensity provided by the World Bank (World Bank, 2017). The CDIAC inventory (v2016; https://dx.doi.org/10.3334/CDIAC/ffe.ndp058.2016) allocates estimates of national emissions to a 1º x 1º grid, primarily distributed according to human population density. A thorough assessment of $2\sigma$ uncertainties in the CDIAC spatial allocation of emissions shows considerable spread in regional uncertainties (Andres et al., 2016).

Our study is not intended to be an exhaustive sampling of inventory approaches but serves to demonstrate the utility of continuous high-accuracy observations as a top-down constraint on emissions evaluations. Our inventory list notably does not include emerging spatially resolved global inventories (e.g. Open Data Inventory for Anthropogenic Carbon Dioxide, ODIAC) (Oda et al., 2018) that were not readily available at the time this work was conducted. At 1km x 1km, ODIAC does have a high spatial resolution of nightlight proxy-based emissions; while this is a valuable method for regions in Europe and North America for example, it is less valuable for China where it is analogous to the CDIAC population-based proxy. In China, power plant emissions are typically located far from end-use regions and the night-light proxy can often break down (Wang, R. et al., 2013). Furthermore, ODIAC power plant emissions use the 2012 Carbon Monitoring for Action (CARMA) database, which notably does not incorporate China-specific power plant data; in these instances, CARMA categorizes China's power plants as "non-disclosed plants" and reports using estimates derived from statistical models using averaged emissions factors – comparable to methods in global inventories subset over China (Ummel, 2012). One of our main goals is to quantify model-observation mismatch associated with use of China-specific power plant data, and ODIAC does not address that issue particularly differently from other global emissions inventories subset over China. For completeness, however, evaluation of global inventories like ODIAC and a suite of increasingly available China-specific inventories (e.g., MEIC) would provide value as part of future model-observation comparison efforts.

Based on multi-year means (2005 to 2009) and 95% confidence intervals derived from two-sample t-tests, we find that within the L_0.90 evaluation region EDGAR and CDIAC report emissions that are significantly lower than ZHAO by typically 20% (-24%, -16%) and 36% (-37%, -34%), respectively. Across China's administrative regions, the highest discrepancy between the global and regional inventories is in Northern China (ZHAO is approximately 30% higher than both EDGAR and CDIAC). In addition, Northern China represents one of the administrative regions with the highest $CO_2$ emissions density (2300 to 3300 Megagrams of $CO_2$ per square kilometer, compared to the average of 700 $MgCO_2$

km$^{-2}$ averaged across China) and is therefore a particularly rich spatial subset for emissions inventory evaluation. A detailed breakdown of emissions by region of China is provided in the SI Table S1. Spatial differences are displayed in SI Figure S7.

Previous work has found that temporal variations in $CO_2$ sources can be significant and surface $CO_2$ can be perturbed from 1.5-8 ppm within source regions based on time of day and/or day of week, resulting from a combination of changes in activity patterns as well as synoptic scale transport effects (Nassar et al., 2013). However, appropriate data for establishing reasonable temporal scaling factors for data-sparse regions such as China are difficult to obtain, and as in the case of Nassar et al. (2013) China's activity factors are based on United States activity factors weighted according to China's EDGARv4.2 emissions patterns. We applied the weekly and diurnal Nassar et al. (2013) scaling factors to our emissions, but these did not generate statistically significant differences from the unscaled versions. These statistically insignificant results suggest that a more rigorous set of temporal scaling factors need to be developed for China. CDIAC does provide monthly gridded inventories with seasonality embedded. However, predictions based on that seasonality deviated even further from the observations than predictions based on constant annual emissions. In the CDIAC global dataset, the seasonality in emissions are based upon generalized global activity factors that are not necessarily appropriate for estimating seasonality of human activity in China. Therefore, in this study we do not explicitly consider diel and seasonal variation in anthropogenic $CO_2$ fluxes.

## 3.4 Vegetation Flux Inventory

We prescribe biotic contributions to the $CO_2$ signal by adapting the VPRM model output for the study domain to generate 0.25º x 0.25º gridded estimates of hourly $CO_2$ net ecosystem exchange (*NEE*) from 2005 to 2009. Details of the VPRM model and output for China are presented in Dayalu et al., 2018. The VPRM is driven by 8-day 500m MODIS surface reflectance values and 10-minute averages of WRF downward shortwave radiation and surface temperature fields. The VPRM parameters are calibrated using eddy flux measurements in the study domain representing each ecosystem type classified according to the International Geosphere-Biosphere Programme (IGBP) scheme. Calibration and evaluation eddy-flux data are obtained from FluxNet and ChinaFlux collaborators. The L_0.90 region is dominated by croplands (Figure S8), in particular the winter wheat and corn dual cropping that characterizes the North China Plain (Dayalu et al., 2018). We use one biosphere model in this study to simplify our assessment of variations across the different emissions inventories. Our selection of the VPRM in particular is based on results from Dayalu et al. (2018), where the VPRM was shown to have significantly lower regional bias than an ensemble of global 3-hourly flux products subset over China.

## 3.5 Background Concentrations

Appropriate quantification of background $CO_2$ concentrations (i.e., the $CO_2$ concentration at the lateral edges of the model domain and/or prior to interaction with domain surface processes) enables realistic

assessment of the study domain's contribution to atmospheric $CO_2$ at varying timescales. CT2015
estimates of $CO_2$ concentrations are provided on a 3° x 2° grid at upwind background locations.
Background values are selected and corrected for large-scale biases using methodology similar to
Karion et al. (2016) where a particle must originate from the outermost domain edge and/or 3000 masl;
further details are provided in the SI Sect. S4. The predicted background $CO_2$ is shown together with
observed $CO_2$ at Miyun for the 1100h-1600h period over the 5-year observational record Figure 3a. For
most of the year the measured $CO_2$ shows large enhancements above background and only in mid-
summer is there a small depletion relative to background values.

**3.6 Quantifying Regional Changes to Background $CO_2$ Concentrations: $\Delta CO_2$**

We define hourly $\Delta CO_2$ as a regional change (enhancement or depletion) imparted to concentrations of
$CO_2$ advected from the boundary ($CO_{2,CT2015}$) such that for each observation hour $\Delta CO_{2,obs}$:

$$\Delta CO_{2,obs} = CO_{2,obs} - CO_{2,CT2015} \tag{1}$$

For each modeled hour $\Delta CO_{2,mod}$, $i$ and $j$ represent the surface gridcell locations and $h$ represents the
hour of the 7-day back trajectory:

$$\Delta CO_{2,mod} = \sum_{0h}^{-168h} \sum_{ij} foot_{ij} \times (ANTH_{ij} + VPRM_{ij}) \tag{2}$$

Note that for the modeled enhancement or depletion, only the VPRM fluxes change hourly; as stated
previously, the annual anthropogenic fluxes are atemporal.

Without a sufficiently dense network of high temporal resolution observations, full-scale inverse
modeling approach to inventory scaling is inappropriate. At annual timescales, where anthropogenic
sources dominate the $CO_2$ signal, we compare annual observed and modeled $\Delta CO_2$ to define a mean
bias and derive a scale factor to quantify the model-observation mismatch based on the slope of the
comparison. Isotopic analysis of atmospheric $CO_2$ from a site in Beijing in 2014 suggests that annually
the fossil fuel burning does dominate the region, contributing 75±15% to the annual signal (Niu et al.,
2016). Annually, the biospheric impact in the region is not zero; rather, the anthropogenic signal
dominates. The biospheric quantity of relevance annually is the net carbon flux as a balance of GPP and
respiration, and is highly uncertain in both sign and magnitude in this region (Piao et al., 2009). In the
Piao et al. (2009) study, regional inversions are based on the very limited dataset of nine sites across all
of Asia. Our assumption of dominant anthropogenic influence in northern china is in keeping with the

priors and process-based models from the relevant regions in Piao et al. (2009) that assume zero and are not significantly corrected by relatively poorly constrained inversions. At seasonal timescales, we use the difference between observed and modeled $\Delta CO_2$ normalized by L_0.90 area to obtain a mass flux offset that combines vegetation and anthropogenic inventories. With the available data it is not possible to independently evaluate both the anthropogenic and biogenic $CO_2$ fluxes. For further details of the scaling technique, please refer to SI Sect. S5.

### 3.6.1 Uncertainty Analysis

The sources of uncertainty in calculations of $\Delta CO_2$ include uncertainty in CT2015 background concentrations, $CO_2$ observations, STILT footprints, anthropogenic inventories, and the biogenic $CO_2$ fluxes from the VPRM. We obtain 95% confidence bounds for $\Delta CO_2$ by following a procedure similar to McKain et al. (2015) and Sargent et al. (2018) that involves bootstrapping daily averages of hourly afternoon values. For monthly and seasonal timescales, we obtain 95% confidence intervals for $\Delta CO_{2,obs}$ by performing a bootstrap on probability distributions of errors in both the CT2015 and observations 1000 times. (See SI Sect. S4 and Figure S9 for details on parameterizing CT2015 uncertainty.) The relevant quantiles are obtained from the resulting distribution, and are reported relative to the mean $\Delta CO_{2,obs}$ of the original data subset. We follow a slightly modified approach for $\Delta CO_{2,mod}$ in that we construct monthly and seasonal residual pools from daily averages of hourly afternoon $CO_{2,mod}$-$CO_{2,obs}$. The residuals—the deviation of the model from the true observed values—represent the total uncertainty in the model and therefore aggregates the effects of uncertainty in the footprints, background, and inventories. Monthly and seasonal 95% confidence intervals of $CO_{2,mod}$-$CO_{2,obs}$ are then obtained from the distribution of bootstrapping the residual pools 1000 times. We then obtain the mean and 95% confidence interval of $\Delta CO_{2,mod}$ by applying the relevant quantiles of the residuals to the mean $\Delta CO_{2,obs}$ of the original data subset. Similar to Sargent et al. (2018) and McKain et al. (2015), distributions of seasonal averages obtained from the above method are used to estimate annual averages and 95% confidence intervals.

Sargent et al. (2018) note that applying the same meteorological model over a long time period (15 months) allows for detection of trends in transport uncertainty. In this study, the drawback of a single location is offset somewhat by a much longer time series (60 months). Absent a dense network of observations, a more sophisticated and extensive error analysis cannot be conducted with meaningful results. Turnbull et al. (2011) faced a similar issue, where weekly flask data collected between 2004 and 2010 from two sites in the NOAA ESRL/WMO sampling network were used to evaluate a bottom-up fossil inventory based on CDIAC and EDGAR estimates. Turnbull et al. (2011) noted the difficulty in assessing the transport error given the paucity of regional observations but also demonstrate the power of top-down assessments given improvements in regional transport modeling and density of observations.

# 4 Results & Discussion

## 4.1 Impact of Seasonality on Evaluation Region

As shown in Figure 2, we find strong seasonality in the footprint percentile contours, in agreement with previous analysis of Miyun observations by Wang et al. (2010). At annual timescales, the L_0.90 region is comparable to the WRF d02 extent. Northern China, including Inner Mongolia, dominate the L_0.90 region both seasonally and annually. Due to the heavy biosphere influence in the regional growing season, previous work by Wang et al. (2010) used Miyun non-growing season measurements of $CO_2$ and carbon monoxide (CO) as an anthropogenic tracer to estimate combustion efficiency for China. When compared to bottom-up estimates of national combustion efficiency, observations suggested 25% higher combustion efficiency than bottom-up estimates of national combustion efficiency; however, Wang et al. (2010) note that the regional (Northern China) and seasonal (winter) subsets could contribute to such a discrepancy. The seasonality exhibited in Figure 2 indeed suggests that combustion efficiency estimates derived from non-growing season measurements alone do not represent anthropogenic processes in provinces south of Miyun that are visible in the observations primarily during the growing season. Low emitting regions northwest of Miyun such as Inner Mongolia influence the site more in the fall and winter relative to other seasons. In the spring and summer, higher emitting regions in provinces south of Miyun are more influential. However, non-growing season $CO_2$ is influenced by often inefficient district heating in the northwest. And, while growing season $CO_2$ is influenced by intense urban activities from Beijing and other cities to the south, vegetation draws down both background and locally-observed $CO_2$ significantly (Figure 3a).

## 4.2 Unscaled Models: Performance at multiple timescales

**Table 1.** Quantification of model-observation mismatch at hourly timescales averaged over 2005-2009 and pooled by season (W=Winter; Sp=Spring; Su = Summer; F = Fall). We provide Standard Major Axis (SMA) slopes and 95% confidence intervals; $R^2$ quantities (those > 0.2 are in bold); and mean bias and root mean square error (RMSE) in ppm.

| | SMA Slope (95%CI) | | | | |
|---|---|---|---|---|---|
| | *All* | *W (JFM)* | *Sp (AMJ)* | *Su (JAS)* | *F (OND)* |
| $\Delta CO_{2,ZHAO+VPRM}$ | 0.89 (0.88,0.91) | 1.0 (1.0,1.1) | 0.74 (0.72,0.77) | 0.88 (0.84,0.92) | 0.92 (0.90,0.95) |
| $\Delta CO_{2,EDGAR+VPRM}$ | 0.77 (0.76, 0.78) | 0.83 (0.81, 0.86) | 0.62 (0.60, 0.65) | 0.83 (0.80, 0.87) | 0.77 (0.74, 0.79) |
| $\Delta CO_{2,CDIAC+VPRM}$ | 0.63 (0.62, 0.64) | 0.63 (0.62, 0.65) | 0.48 (0.46, 0.50) | 0.79 (0.75, 0.82) | 0.56 (0.54, 0.58) |
| | $R^2$ | | | | |
| | *All* | *W (JFM)* | *Sp (AMJ)* | *Su (JAS)* | *F (OND)* |
| $\Delta CO_{2,ZHAO+VPRM}$ | **0.49** | **0.56** | **0.26** | **0.22** | **0.56** |
| $\Delta CO_{2,EDGAR+VPRM}$ | **0.47** | **0.55** | **0.21** | 0.18 | **0.55** |

| $\Delta CO_{2,CDIAC+VPRM}$ | **0.43** | **0.55** | 0.17 | 0.13 | **0.54** |
|---|---|---|---|---|---|
| | **Mean Bias (RMSE), ppm** | | | | |
| | *All* | *W (JFM)* | *Sp (AMJ)* | *Su (JAS)* | *F (OND)* |
| $\Delta CO_{2,ZHAO+VPRM}$ | 0.32 (9.2) | 0.014 (7.9) | -0.033 (8.3) | 3.1 (11) | -1.1 (9.7) |
| $\Delta CO_{2,EDGAR+VPRM}$ | -2.0 (9.3) | -2.2 (7.7) | -1.9 (8.7) | 0.25 (10.8) | -3.4 (10.1) |
| $\Delta CO_{2,CDIAC+VPRM}$ | -3.3 (9.9) | -3.1 (8.1) | -3.3 (9.2) | -1.1 (11.3) | -5.0 (11.1) |

We evaluate unscaled model performance relative to observations at hourly, seasonal, and annual timescales. While inventory scaling is performed at the policy relevant scales of seasons and years, examination of the models at shorter timescales provides insight into model bias and error aggregation at longer timescales. Table 1 summarizes hourly model bias across all years and pooled by season.

All modeled hourly quantities include the same biological component from VPRM, background concentrations, and transport model such that the only source of variation among models is the anthropogenic inventory. With a few exceptions that are discussed in the following sections, $CO_{2,EDGAR+VPRM}$, $CO_{2,CDIAC+VPRM}$, $\Delta CO_{2,EDGAR+VPRM}$, and $\Delta CO_{2,CDIAC+VPRM}$ systematically underestimate observations as indicated by larger deviation below the 1:1 line in the comparison of modeled to

measured $\Delta CO_2$ (Table 1, Figure 3b-d.)

### 4.2.1 Hourly

We examine the distribution of modeled-measured residuals at hourly timescales for each anthropogenic

inventory. While standard deviations are consistent across all models of $CO_2$ flux ($1\sigma$=9ppm; Figure 3.e-g) $\Delta CO_{2,ZHAO+VPRM}$ exhibits the least bias relative to observations with a mean residual of 0.32(0.12,0.53) ppm. In contrast, $\Delta CO_{2,EDGAR+VPRM}$ and $\Delta CO_{2,CDIAC+VPRM}$ display significantly greater bias by typically underestimating observations by large amounts: -2.0(-1.8,-2.2) ppm and -3.3(-3.1,-3.5) ppm, respectively. Here, the 95% confidence intervals are derived from a two-sample t-test. The

EDGAR and CDIAC underestimation of $\Delta CO_2$ at the hourly scale is consistent across longer timescales of seasons and years as discussed in the following sections, but we note where there are likely aliased effects of the uncertainty in the VPRM biogenic component.

### 4.2.2 Seasonal

The seasonally averaged modeled and measured $\Delta CO_2$ values shown in Figure 4 illustrate the overall

biases for the four inventories. Outside of June, July, August, and September, the anthropogenic signal dominates in northern China (Wang et al., 2010). We see from Table 1 that during seasons where biological activity is lower or significantly lower than anthropogenic activity, there is a consistent

discrepancy among the $CO_2$ modeled by the three different anthropogenic inventories suggesting systematic differences largely attributable to the anthropogenic component (as we do not vary any other component) . In the fall, where respiration is the dominant biological process, all three modeled


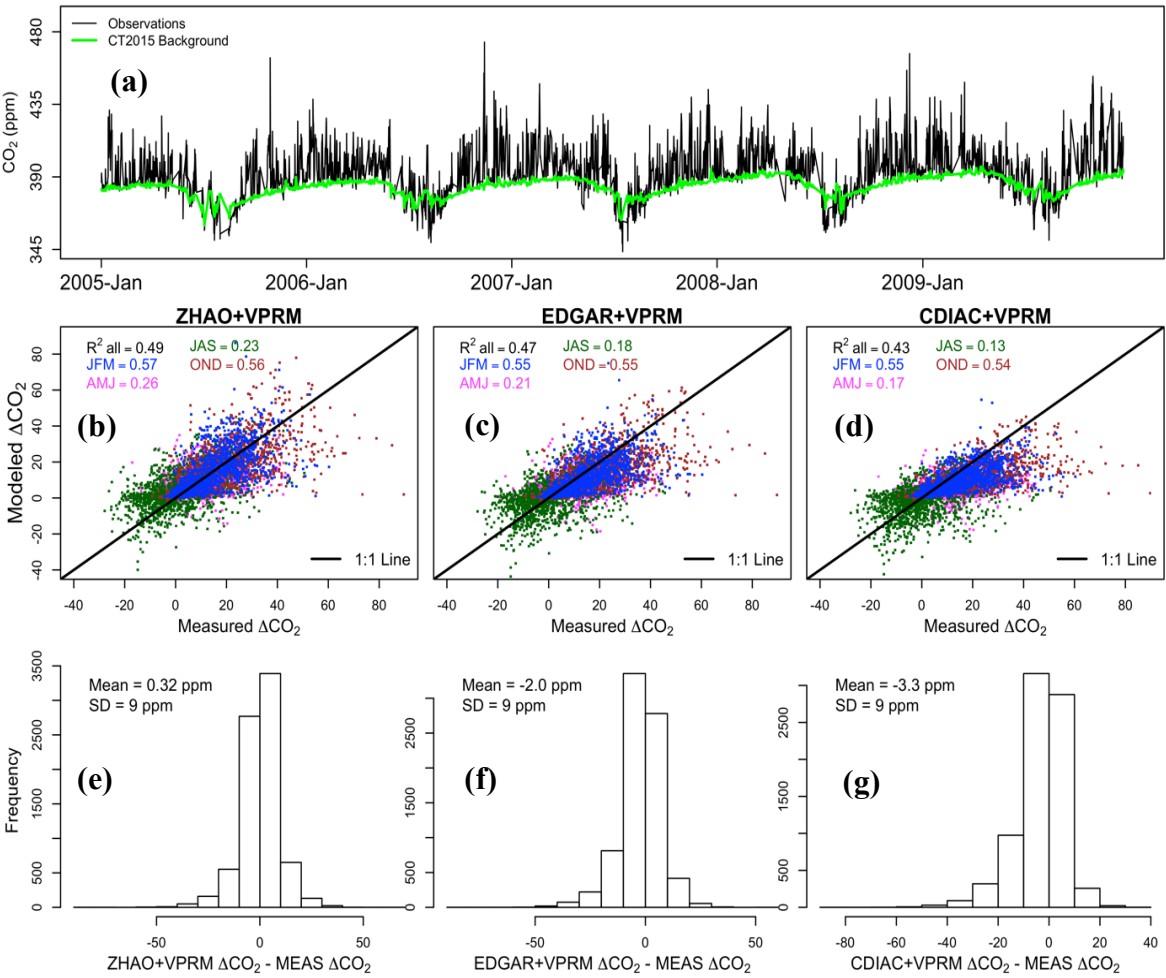

**Figure 3.** Hourly (1100 to 1600 Local Time) Modeled and Measured $CO_2$ and $\Delta CO_2$. Measured $CO_2$ and modeled CT2015 background concentrations are displayed in (a). Modeled versus measured $\Delta CO_2$ for each anthropogenic inventory is shown in (b)-(d), colored by season. Histograms of modeled-measured residuals are shown in (e)-(g). The VPRM vegetation component is included in all modeled $\Delta CO_2$ values.

quantities are consistently lower than observations—a likely a consequence of the known underestimate of ecosystem respiration by the VPRM (Dayalu et al., 2018). Even so, China's significant anthropogenic

component still dominates during these months. During the winter season, where all biospheric activity is at a minimum, the model-observation mismatch is most reflective of biases among anthropogenic inventories rather than aliased impacts from the VPRM. As shown in the winter data in Table 1, ZHAO displays the least bias relative to observations (0.01ppm) followed by EDGAR(-2.2ppm) and CDIAC (-3.1ppm).

With the exception of the peak JAS growing season, $\Delta CO_{2,EDGAR+VPRM}$ and $\Delta CO_{2,CDIAC+VPRM}$ typically underestimate $\Delta CO_{2,OBS}$, even within the 95% uncertainty bounds. The VPRM has a limited calibration network that contributes to an underestimate of regional $CO_2$ drawdown during the growing season (Dayalu et al., 2018). Therefore, while $\Delta CO_{2,ZHAO+VPRM}$ agrees within 95% confidence bounds with $\Delta CO_{2,OBS}$ during the non-growing seasons, $\Delta CO_{2,ZHAO+VPRM}$ generally overestimates $CO_2$ concentrations in the growing season (Figure 4a). $\Delta CO_{2,EDGAR+VPRM}$ (Figure 4b) and $\Delta CO_{2,CDIAC+VPRM}$ (Figure 4c) display lower $CO_2$ concentrations and generally result in better agreement with observations during the peak growing season than at other times of the year; however, our wintertime and overall analysis at hourly timescales (Figure 4, Table 1) suggests this is an artifact of lower anthropogenic emissions

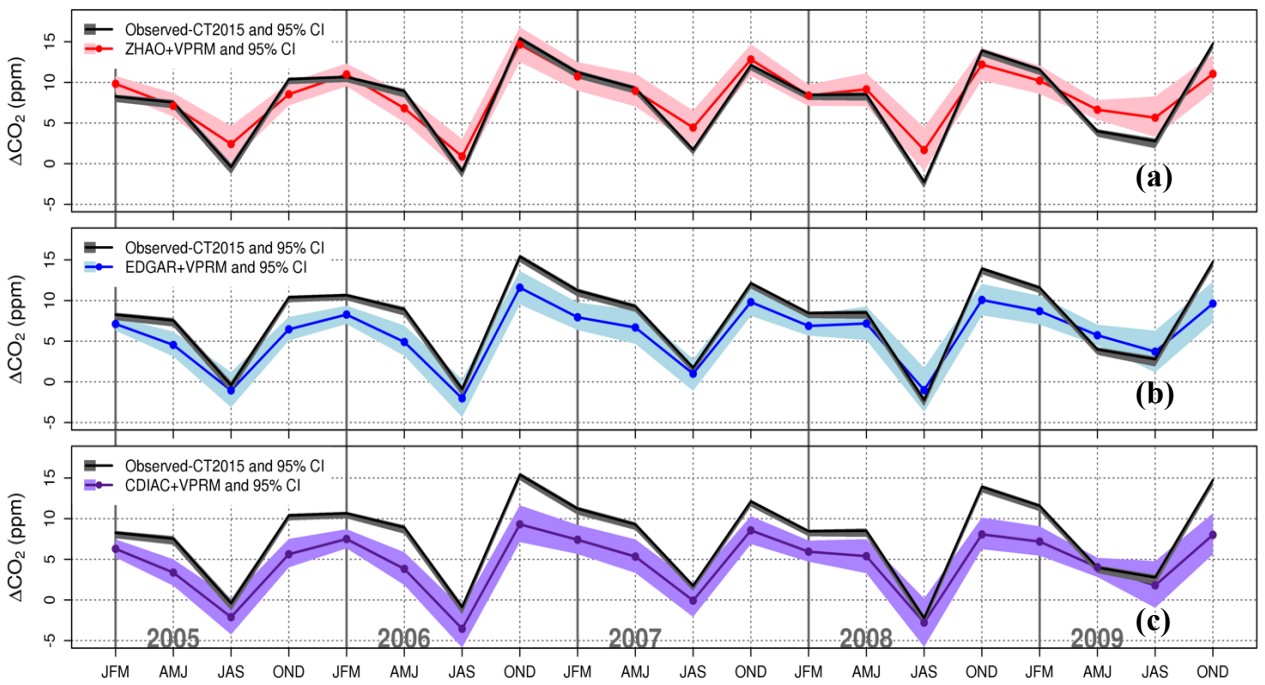

**Figure 4.** Modeled and Measured Seasonal $\Delta CO_2$. CT2015 background is subtracted from observations to provide observed $\Delta CO_2$ (black line). 95% confidence bounds are derived from bootstrapping hourly afternoon concentrations for each season.

estimates relative to ZHAO that counteracts the VPRM underestimating drawdown. Even during the growing season, $\Delta CO_{2,CDIAC+VPRM}$ agrees with observations typically at its upper confidence limits. However, during times of the year where the impacts of underestimated respiration become more significant (e.g., Fall) it is possible that the seemingly better agreement of ZHAO+VPRM is linked to a counteracting effect of overestimated anthropogenic emissions.


As ZHAO+VPRM demonstrates the least bias relative to observations at hourly and seasonal scales, we model the relative contributions to the monthly signal during the May through September peak regional growing season as defined by Wang et al. (2010). Figure 5 displays the results from partitioning the mean monthly $\Delta CO_{2,ZHAO+VPRM}$ signal as a multi-year average into anthropogenic and vegetation contributions. While the WRF-STILT-VPRM framework has been successfully adapted for similar $CO_2$ inventory evaluation studies in North American regions where biogenic fluxes dominate surface processes (Karion et al., 2016; Matross et al., 2006), Figure 5 shows the relative magnitude of biogenic fluxes and anthropogenic emissions in the Northern China region is comparable during peak summer, making it difficult to independently constrain them with observational data. As noted in Sect. 3, the regional peak uptake during the growing season occurs with the onset of the corn growing season around July and August. The atypical lower uptake during June represents the winter wheat/corn transition period. These results are consistent with the biological component estimated by Turnbull et al. (2011). Furthermore, knowledge of the relative contribution of vegetation and anthropogenic processes




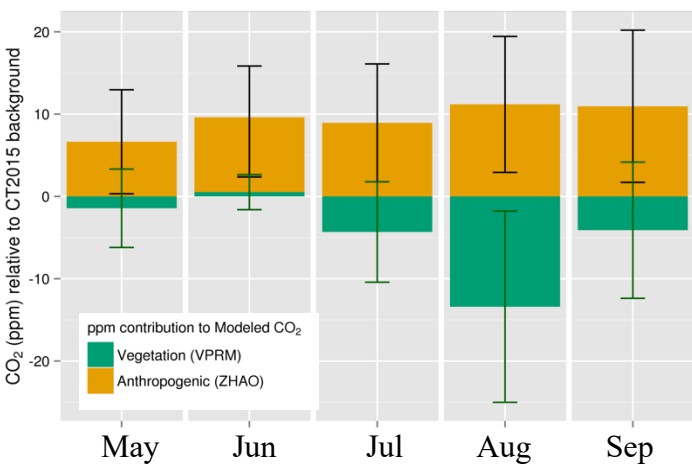

**Figure 5.** Modeled mean monthly contribution (ppm) to Miyun $CO_2$ concentrations from vegetation (VPRM) and anthropogenic (ZHAO) sources. Enhancement and depletion are relative to advected CT2015 background concentrations during the regional growing season (MJJAS), averaged over 2005 to 2009. Vertical lines represent 1-σ of monthly averages (Green: Vegetation; Black: Anthropogenic). Negative values represent depletion from CT2015 background; positive values represent enhancement of CT2015 background.

to the $CO_2$ signal during the peak growing season is necessary to interpret satellite retrievals of $CO_2$ over the region (Dayalu et al., 2018).

### 4.2.3 Annual

Aggregation of uncertainty and anthropogenic inventory biases at shorter timescales becomes most apparent at the annual timescales. For annual budgeting we follow the assumptions of Piao et al. (2009) and Jiang et al. (2016) that agricultural systems are in annual carbon balance because crop biomass has a short residence time. In the absence of data on regional transfer of agricultural products and proportion of grains used in situ for livestock vs. human consumption in China this is the most
conservative assumption to make. Given the dense population in most of Beijing province we expect there may be net import of agricultural products from outside the L_0.90 region, which would show up as additional respiration not captured by VPRM, but that term will be small relative to the anthropogenic $CO_2$ (Figure 5) (Dayalu et al., 2018). Therefore, while the VPRM is implicitly included in the modeled annual $CO_2$ and $\Delta CO_2$, vegetation carbon stocks (including harvested products and crop
residues) portions of the L_0.90 region with widespread agriculture largely turn over such that only the anthropogenic inventories dominate the modeled $CO_2$ signal. We evaluate annual $CO_2$ including

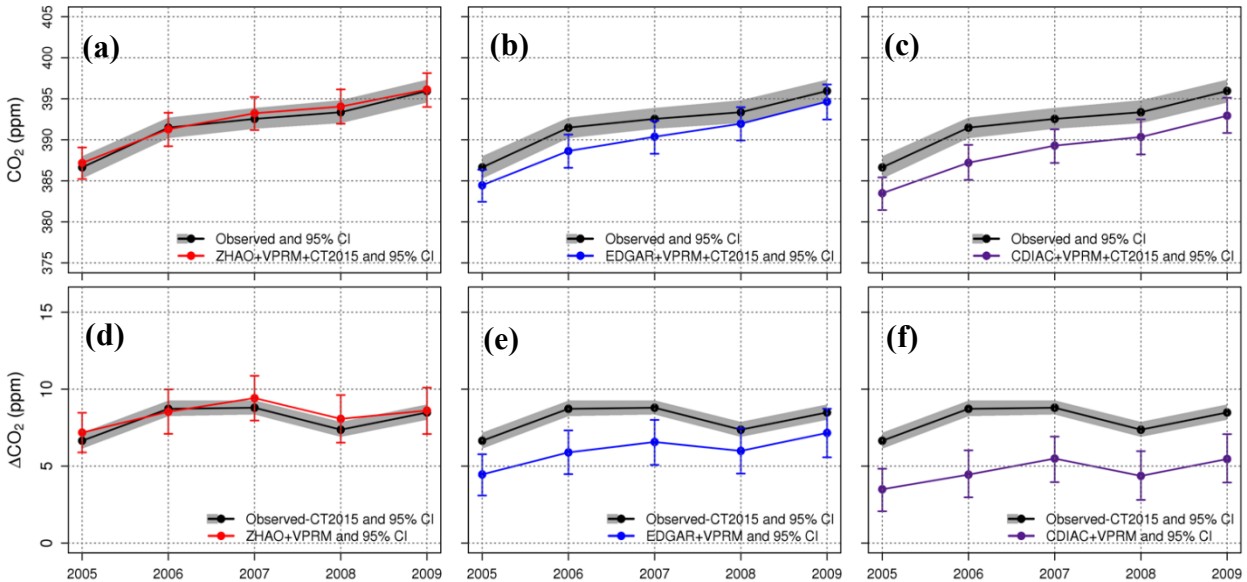

**Figure 6. Mean annual $CO_2$ and $\Delta CO_2$ over entire study time period.** (a-c) $CO_2$ annual concentration; (d-f) $\Delta CO_2$ (regional enhancement, after removal of advected CT2015 background) with bootstrapped 95% confidence intervals.

CT2015 background (Figure 6a-c) and as regional enhancement relative to background (Figure 6d-f). We show that for all years, $CO_{2,ZHAO+VPRM}$ and $\Delta CO_{2,ZHAO+VPRM}$ agree tightly within 95% uncertainty to observations (Figure 6a, Figure 6d). EDGAR+VPRM and CDIAC+VPRM are consistently biased significantly lower than observations.

## 4.3 Evaluation of inventories at seasonal and annual timescales

We quantify model-observation mismatch by estimating the additive flux corrections at seasonal timescales and multiplicative corrections at annual timescales. We emphasize that these "corrections", or scalings, are not optimizations; rather, they simply reflect the extent to which the individual anthropogenic+VPRM flux models deviate from the observations. Complete seasonal and annual scaling results are provided in the SI Sect. S5, and Tables S2-S3.

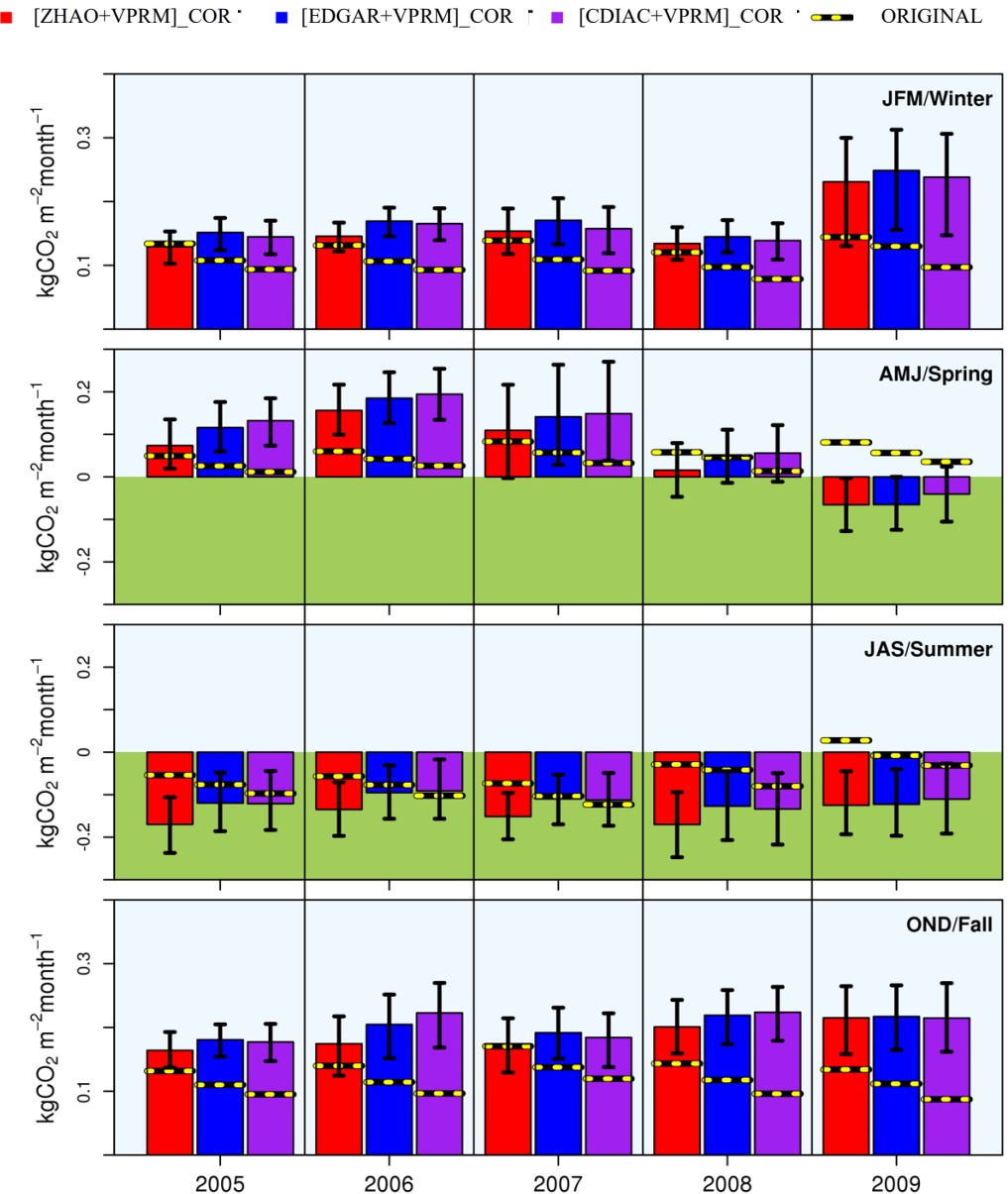

**Figure 7.** Scaled Seasonal Fluxes in the L_0.90 region (kg $CO_2$ m$^{-2}$ month$^{-1}$). Anthropogenic and vegetation inventories are scaled together ([ANTH+VPRM_COR]). Black and yellow dashed line is the seasonal flux estimated by the original ANTH+VPRM model. All models have the same vegetation component (VPRM) and differ only in the anthropogenic inventory source. Shaded green represents negative flux (uptake by biosphere). Scaling based on additive corrections; difference among scaled inventories is due to differing spatial allocations by anthropogenic inventories. Bootstrapped 95% confidence intervals are represented by the black vertical lines.

The observational record informing the scaling integrates the biological and anthropogenic signals. At the seasonal scale, where biological processes are significant contributors to the signal, we scale the sum of the anthropogenic and biological fluxes (Figure 7). Scaled non-growing season flux estimates


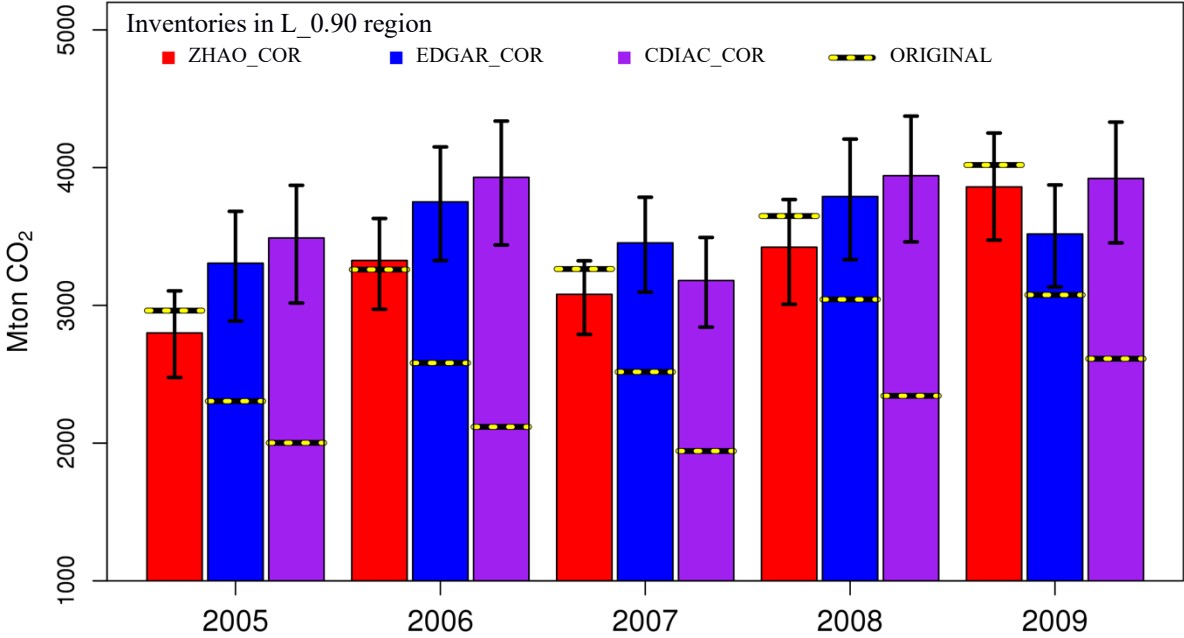

**Figure 8.** Annually scaled emissions in L_0.90 region. Scaling is based on multiplicative scaling factors. Difference among scaled inventory means is due to differing spatial allocations in original anthropogenic inventories. Bootstrapped 95% confidence intervals are represented by the black vertical lines. *Note the y-axis origin begins at 1000 Mton $CO_2$ for visual clarity.*

are

higher than unscaled values, partially accounting for the VPRM generally underestimating ecosystem respiration by an additive offset throughout the year (Dayalu et al., 2018). The multi-year seasonal results in Table 1 suggest that this offset can aggregate to a 1-2ppm difference; the result would be a
shift in baseline rather than overall pattern for each of the three simulations. As the vegetation and all other components are controlled across models, the inter-model variance reflects the relative performance of the anthropogenic estimates. We find that in the non-growing months the original ZHAO+VPRM inventory typically remains within the 95% confidence bounds of the scaled inventory. However, both EDGAR+VPRM and CDIAC+VPRM are consistently significantly lower than their
scaled counterparts. At least in the winter, where biogenic processes are at a minimum, this suggests that both EDGAR and CDIAC underestimate anthropogenic emissions, and that ZHAO estimates are closer to actual emissions. Improved representation of temporal anthropogenic activity factors and biosphere processes are needed to extend the conclusions of anthropogenic inventory performance to all

seasons. In the absence of such data, it is not possible to conclusively state whether model-data
mismatch is rooted in anthropogenic emissions biases or biogenic biases.   During the growing seasons,
however, the afternoon vegetation signal is significant, and the picture is more complex. In the spring,
the $CO_2$ signal at Miyun is significantly affected by the North China Plain winter wheat growing season.
The effect of scaling in the spring from 2005 to 2007 is to increase $CO_2$ emissions with a net positive
seasonal flux; however, in 2008 and 2009 we find the net seasonal flux becomes negative such that
uptake dominates emissions. The prior models in all cases predict positive flux. During the summer
months, ZHAO+VPRM predicts more emissions and/or less uptake relative to EDGAR+VPRM and
CDIAC+VPRM. Scaling of summertime fluxes serves to significantly increase ZHAO+VPRM uptake
estimates; the EDGAR+VPRM and CDIAC+VPRM prior estimates are within the 95% confidence
bounds of the scaling for reasons discussed previously.

**Table 2.** Annual scaling factors (95% CI) and corresponding corrected emissions for L_0.90 inventory evaluation region.

| | | Scaling Factor (95% CI) | Corrected Emissions, $MtCO_2$ (95% CI) | Original emissions, $MtCO_2$ |
|---|---|---|---|---|
| 2005 | ZHAO | 0.95 (0.84, 1.0) | 2800 (2476, 3105) | 3015 |
| | EDGAR | 1.4 (1.3, 1.6) | 3306 (2886, 3683) | 2322 |
| | CDIAC | 1.7 (1.5, 1.9) | 3489 (3017, 3871) | 1930 |
| 2006 | ZHAO | 1.0 (0.91, 1.1) | 3326 (2972, 3631) | 3273 |
| | EDGAR | 1.5 (1.3, 1.6) | 3751 (3325, 4150) | 2586 |
| | CDIAC | 1.9 (1.6, 2.0) | 3930 (3438, 4338) | 2160 |
| 2007 | ZHAO | 0.94 (0.85, 1.0) | 3080 (2789, 3324) | 3588 |
| | EDGAR | 1.4 (1.2, 1.5) | 3454 (3096, 3785) | 2799 |
| | CDIAC | 1.6 (1.5, 1.8) | 3180 (2842, 3493) | 2260 |
| 2008 | ZHAO | 0.94 (0.82, 1.0) | 3422 (3008, 3768) | 3685 |
| | EDGAR | 1.2 (1.1, 1.4) | 3790 (3332, 4207) | 3095 |
| | CDIAC | 1.7 (1.5, 1.9) | 3941 (3461, 4374) | 2395 |
| 2009 | ZHAO | 0.96 (0.86, 1.1) | 3860 (3474, 4251) | 3974 |
| | EDGAR | 1.1 (1.0, 1.3) | 3518 (3133, 3874) | 3298 |
| | CDIAC | 1.5 (1.3, 1.7) | 3921 (3454, 4330) | 2543 |


We report annual scaled anthropogenic inventories in the L_0.90 region in Fig. 8 and Table 2 as
$MtCO_2yr^{-1}$. As discussed previously, the annual scalings are applied only to the anthropogenic
inventory, as the signal at the annual timescale is effectively dominated by anthropogenic emissions; net
ecosystem fluxes are expected to be relatively minor in the L_0.90 region in comparison. For all years,
the emissions estimated by the original ZHAO inventory lie within the 95% confidence bounds of the
scaled ZHAO inventory. However, for EDGAR and CDIAC, the original inventories consistently
underestimate observations. Averaged over the five-year study period, EDGAR and CDIAC lead to
modeled estimates of $CO_2$ mixing ratios that are typically lower than observations by 30% and 70%

respectively (Fig. 6). Averaged across the five years, this translates to EDGAR and CDIAC being scaled relative to their unscaled values in the L_0.90 region by 1.3 and 1.7, respectively (Fig. 8; Table 2). In the case of EDGAR, we note a general increase in observational agreement from 2005 to 2009.

## 4.4 Potential Contributions to Regional Carbon Emissions Patterns from 2005 to 2009

We examine the statistical significance of the inter-annual observed concentration and enhancement differences using a two-sample t-test (Table 3). The observed concentrations including advected global background (Figure 6, top row) display an overall increasing trend of 1.87 (1.8, 1.9) ppm $CO_2$ yr$^{-1}$
between 2005 and 2009, in agreement with flask samples obtained from nearby WMO sites between 2007 and 2010 (Liu et al., 2014). The inter-annual increases are statistically significant (Table 3). However, when we remove the modeled background to more closely examine regional patterns that would otherwise be drowned out by the global signal, we find that the regional $\Delta CO_2$ trend (Figure 6, bottom row; Table 3) does not parallel the increasing global $CO_2$ trend (Figure 6 top row; Table 3).
Regionally, the observed enhancements increase from 2005 to 2006 and plateau in 2007 before decreasing in 2008. Regional $\Delta CO_2$ increases again in 2009. Earlier work by Wang et al. (2010) extended the Miyun observations of $CO_2$ growth rate to all of China and estimates a lower $CO_2$ growth rate than previously suggested. However, Figure S6 suggests local reductions in regions influencing Miyun, possibly in preparation for the Beijing Olympics, are partially offset by increases elsewhere. A
larger network of sites would be needed to quantify this further in order to evaluate the $CO_2$ growth rate for other regions in China and for China as a whole.

In Figure 9a we estimate Gross Regional Product (GRP) for eight of China's 34 provincial-level administrative units, specifically those encompassed significantly by the L_0.90 region: Beijing,
Tianjin, Henan, Shanxi, Shandong, Hebei, Inner Mongolia, and Liaoning. Using data from the International Monetary Fund (IMF; https://www.imf.org/en/Data) and World Bank (World Bank, 2017, we retrieved the GDP for each of the above provinces and summed them to estimate the GRP. GDP calculations are inherently uncertain and were available as single values for each province per year. A more extensive economic analysis to estimate uncertainty of these values is beyond the scope of this
study. Key economic events occurred during the study time period and are likely contributors to the observed interannual variation in regional $CO_2$ emissions (Figure 6d-e) and a doubling of GRP from 2005 to 2009 (Figure 9a). In particular, the time period from 2005-2009 saw industrial energy efficiency improvements beginning in 2007 under the 11[th] FYP; preparations for and staging of the 2008 Beijing Summer Olympics; the global financial crisis in late 2008; and a large Chinese fiscal stimulus in 2009.
We further note that the global financial crisis of 2008 correlates with a plateauing of the percentage contribution of northern China GRP to national GDP (Figure 9a).

**Table 3.** Inter-annual observed $CO_2$ and $\Delta CO_2$ differences. Differences are of observations between consecutive years. 95% confidence intervals are derived from a two-sample t-test. Italicized entries denote instances where the inter-annual difference is not statistically significant (confidence interval includes zero).

| Time Interval (y₂-y₁) | $CO_{2,OBS}$ (ppm) Mean Difference (95% CI) | $\Delta CO_{2,OBS}$ (ppm) Mean Difference (95% CI) |
|---|---|---|
| **2006-2005** | 4.86 (4.5, 5.2) | 2.08 (1.9, 2.3) |
| **2007-2006** | 1.08 (0.69, 1.5) | *0.0693 (-0.15, 0.29)* |
| **2008-2007** | 0.772 (0.37, 1.2) | -1.43 (-1.6, -1.2) |
| **2009-2008** | 2.60 (2.2, 3.0) | 1.12 (0.88, 1.4) |
| **2009-2005** | 9.31 (8.9, 9.7) | 1.84 (1.6, 2.0) |

As policy targets are often measured as relative changes over multiple years, an important component of emissions inventories is their ability to accurately capture multi-year changes. Observations indicate enhancements above background $CO_2$ increased by 28% (22%, 34%) between 2005 and 2009. ZHAO+VPRM estimates a 20% increase over the same time period while EDGAR+VPRM and CDIAC+VPRM estimate 61% and 56% increases respectively.

## 4.5 Implications for Assessing National Carbon Emission Targets

China has pledged a 60-65% reduction in carbon intensity by 2030 and has additionally set a benchmark of 40-45% reduction in carbon intensity by 2020, where both targets are relative to the baseline year 2005 (NDRC, 2015; Guan et al., 2014). However, Guan et al. (2014) found that provincial trends in carbon intensity can vary significantly from national trends. Using the GRP values shown in Figure 9a, we calculate a Northern China regional carbon intensity incorporating the eight provinces encompassed significantly by the L_0.90 region (Figure 9c). We also estimate an L_0.90 regional carbon intensity based on the official national energy-related $CO_2$ emissions in NDRC (2015); we scale the national total by 39% (35%,42%) which is the mean (range) contribution of the L_0.90 region to the national emissions in 2005, averaged across the three unscaled gridded emissions inventories. We emphasize that carbon intensity values are inherently uncertain due to complexities in GRP and Gross Domestic Product (GDP) calculations such as double-counting due to inter-provincial trade or spatial mismatch between emissions and economic data. Nevertheless, the analysis provides valuable insight into trends rather than precise values.

Over the study time period, the GRP of the L_0.90 region more than doubled (Figure 9a), exhibiting a moderate, positive correlation with the increasing trend in emissions (Figure 9b). Coinciding with the 2008 Beijing Summer Olympics, the region's contribution to China's GDP grew from approximately 13.5% in 2007 to nearly 16% in 2008, representing a 20% increase, before plateauing into 2009 (Figure

9a). As noted in Guan et al. (2014), reductions in carbon emissions intensity can come about via two main pathways: the first, within industries, through increased energy efficiency combined with expanded production capacity; the second, across the economy, through structural shifts from energy-intensive industrial sectors to service sectors. The doubling of GRP with the apparent reduction in regional carbon intensity suggests a combination of enlarged production capacity (including production of higher valued goods) and a shift toward service-oriented economy. In the former instance, a larger production capacity tends to reduce the overall energy (and, therefore, carbon) consumption of a single production unit. In the latter instance, the energy consumption by the service sector is considerably lower than that required by industrial and manufacturing processes. In the northern China region, however, industry continues to dominate the economy suggesting that carbon intensity reductions are more due to enlarged production capacity. From 2005 to 2009, carbon intensity for the L_0.90 region decreased by 47% (28%,65%), based on a one-sample t-test of pooled emissions intensity changes across scaled inventories. Analysis presented by organizations such as the World Bank (World Bank, 2017) suggests China's carbon intensity at the national level decreased by 20% in 2009 relative to 2005. However, we note that the carbon emissions data source for the World Bank carbon intensity calculations is CDIAC. We have shown that at least for the L_0.90 region, CDIAC emissions lead to significant underestimates of observations. Our work here suggests that carbon accounting organizations such as the World Bank would benefit from basing their national estimates for China on a variety of inventories, incorporating increasingly available China-specific approaches (including but not limited to MEIC and PKU), EDGAR, and newer global inventories such as ODIAC. However, we emphasize a crucial point with respect to the value of carbon intensity targets, in agreement with Guan et al. (2014): carbon intensity targets are especially misleading in developing countries where absolute emissions continue to significantly grow in concert with economic development goals. We see that despite the decreasing carbon intensity of the region, pooled emissions estimates from the three scaled inventories suggest an 18% increase in absolute emissions from 2005-2009 (Table 2, Figure 9b). In terms of the climate impact, it is the absolute carbon emissions rather than the carbon intensity that ultimately matters.

705

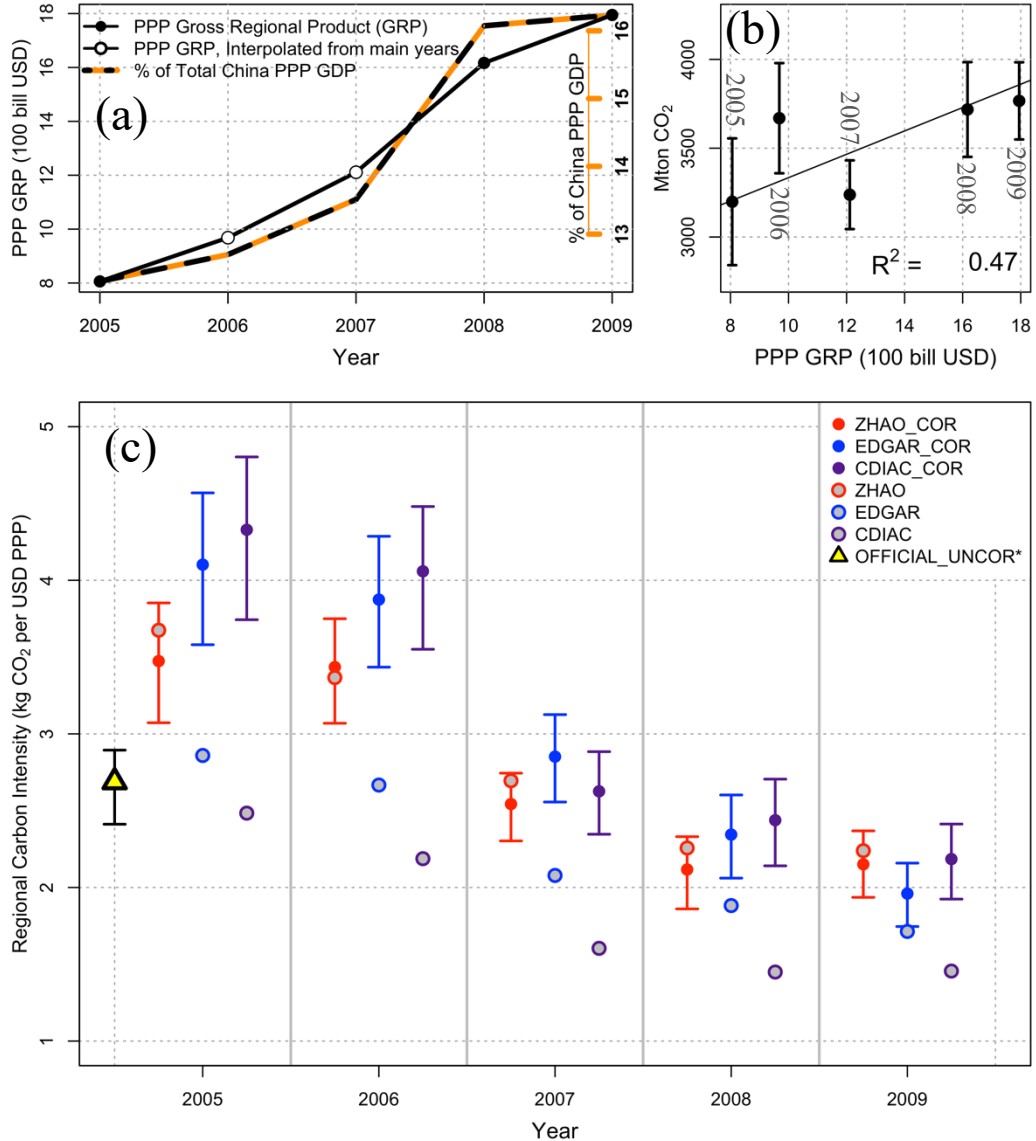

**Figure. 9. Estimates of Regional Carbon Intensity (kg CO₂ USD$_{PPP}^{-1}$).** (a) PPP GRP by year and as a % of China's national GDP. No PPP GRP values were available for 2006 and 2007; PPP GRP for these years was derived from linearly interpolated ratio of Nominal GRP/PPP GRP for 2005, 2008, and 2009. (b) Correlating corrected regional emissions from Table 2 with PPP GRP; values are pooled annual means among ZHAO, EDGAR, and CDIAC with 1-σ error bars. (c) Regional Carbon Intensity using scaled (solid) and unscaled (grey) CO₂ estimates. Error bars are bootstrapped 95% confidence intervals. GRP, GDP data from IMF and World Bank. Provinces used in GRP calculation are those significantly encompassed by L_0.90 region Beijing, Henan, Shanxi, Tianjin, Shandong, Hebei, Inner Mongolia, and Liaoning. *Estimated by scaling the official national emissions total by the average contribution (39%) of L_0.90 region to total emissions in 2005. Uncertainty bars represent the % contribution range estimated by ZHAO, EDGAR, and CDIAC in 2005 (35%, 42%).*

## 5 Conclusions

Continuous hourly $CO_2$ observations, significantly influenced by the heavily $CO_2$-emitting Northern China region, are used in a top-down evaluation and scaling of three bottom-up $CO_2$ flux inventories. We focus on the policy-relevant time interval from 2005 to 2009, noting that 2005 is China's baseline year for carbon commitments. The three inventories are distinct in their anthropogenic component, with a common biogenic flux component provided by the VPRM, a simple satellite data-driven biosphere model calibrated with ground-level ecosystem observations. The ZHAO anthropogenic emissions inventory incorporates a regional approach to China's $CO_2$ emissions estimation, using activity data at the provincial and facility-levels as well as domestic emission factors. The EDGAR and CDIAC emissions inventories incorporate a greater reliance on global averages and China's national statistics and international default emission factors, and depend more heavily on proxies (e.g., population) to allocate the emissions geographically. The three anthropogenic inventories represent a range of methods used to estimate emissions for China.

The Northern China administrative region, excluding Inner Mongolia, dominates the L_0.90 region which is the region over which we distribute the model-observation mismatch (Figure 2). We find strong seasonality in the L_0.90 region, ; the northwest features more strongly in the non-growing season and there is a more symmetric influence in the growing season. Within the L_0.90 region, EDGAR and CDIAC are—on average across the five study years—lower than ZHAO by 20% and 36%, respectively. Across administrative regions, the highest discrepancy between the global and regional inventories is in Northern China, where the ZHAO inventory estimates emissions that are on average 30% higher than both EDGAR and CDIAC (SI, Table S1).

We find the ZHAO+VPRM inventory generally agrees very closely with observations, often significantly better than the nationally referenced inventories at all timescales (hourly through annually), with the exception of the peak growing season. During the peak growing season, the regional enhancement to background $CO_2$ concentrations is modeled as approximately zero, due to an agriculturally dominated vegetation signal that is equal in magnitude and opposite in sign to the anthropogenic signal (Dayalu et al., 2018). While this agrees with previous work by Turnbull et al. (2011), in both that study and the present study the sparse data prevents a more conclusive statement about anthropogenic inventory performance during the regional growing season. At annual timescales, the anthropogenic signal dominates, and we find that emission rates from EDGAR and CDIAC lead to underestimated emissions in the Northern China region by an average of 30% and 70%, respectively, averaged across all study years. We note that the discrepancy between the EDGAR-based timeseries and the observations generally decreases over the five-year study period. In contrast, emission rates from the ZHAO inventory gives *a priori* results very close to observations throughout and is not significantly affected by the scaling: the error bars for the scaled estimates consistently include the original estimate. Note that the EDGAR and CDIAC inventories can differ from -10% to -20% relative to ZHAO in their national emissions totals (Table S1). The inventories evaluated here exhibit distinct differences in their

ability to match observations. However, observational data from a network of sites strategically located in and around the eastern half of China would be required to (1) examine whether differences in spatial allocation approaches contribute to differences among the inventories and (2) conduct actual optimizations of the inventories.

We find that carbon intensity in the region has decreased by 47%(28%, 65%) from 2005 to 2009, from approximately $4kgCO_2/USD_{PPP}$ in 2005 to about $2kgCO_2/USD_{PPP}$ in 2009 (Figure 9c). However, we see that despite the decreasing carbon intensity of the region, there is an 18% increase in absolute emissions over time, affirming the point made by Guan et al. (2014) that meeting carbon intensity targets in emerging economies can be at odds with making real climate progress (Table 2, Figure 9b).


Despite the limitations of having data from a single site, this analysis demonstrates how a long time series of continuous observations can identify apparent overall biases in some inventories. Our results, while specific to northern China regional emissions in particular, also provide some insight into current methods of carbon emissions accounting for China as a whole. We emphasize that this work is intended
to be a comparison of emission rates from a subset of anthropogenic $CO_2$ inventories over northern China that were readily available at the time this research began and is not intended to be an advocate or criticism of any single published inventory. Rather, we use a long 60-month continuous observational record to examine model-data mismatch in an important carbon emitting region where local data is difficult to access and global datasets are forced to rely on the best available public data, which are not
necessarily accurate assumptions of China-specific activity. Second, while we recognize the height limitations –and therefore the footprint—of the Miyun receptor its topographic advantage along with the low-productivity vicinity, make it similar to other short-tower sites suitable for regional analysis. In addition, a detailed assessment of uncertainty stemming from errors in transport, biogenic inventories, and inventory spatial allocation remains a challenge. Independent verification from concurrent aircraft
measurements (for example) or multi-level inlet locations were not available to quantify the impact of absolute and relative inlet location on transport uncertainty. Finally, we emphasize our implied seasonal and annual "corrections", or scalings, of modeled $CO_2$ relative to observations are not optimizations; rather, they simply reflect the extent to which the individual anthropogenic+VPRM $CO_2$ flux models deviate from the observations. At least in the winter, where biogenic processes are at a minimum, the
low bias of ZHAO-modeled $CO_2$ concentrations suggests the ZHAO inventory is closer to actual emissions. However, improved representation of temporal anthropogenic activity factors and biosphere processes are needed to extend the conclusions of anthropogenic inventory performance to all seasons. Effectively evaluating and constraining inventory emissions rates at relevant spatial scales requires multiple stations of high-temporal resolution observations, as well as improvements and greater
diversity in observationally-constrained biogenic flux models. In its current configuration, the single biogenic flux model precludes a comprehensive multi-seasonal and annual disentangling of contributions to $CO_2$; particularly in our annual scale analysis, we are ascribing more uncertainty to the anthropogenic inventories over the biogenic contributions. Absent data from a dense network of ecosystem flux and atmospheric measurements, there will constantly be a tradeoff between drawing

conclusions using low-temporal resolution flask measurements from a few sites and continuous data from a single location.

In situ $CO_2$ observations interpreted within a high-resolution model framework such as described in this study provide a powerful constraint to test and correct spatially explicit inventories. The observation

station available for the 2005-2009 period was strategically located to provide information on one of the highest $CO_2$ emitting regions of China. Within the limitations described above, the observations provide strong evidence supporting the use of China-specific methods, such as those employed in ZHAO, for China's $CO_2$ emissions inventory derivation. In future, access to a spatially dense network of measurements will allow for a sophisticated error analysis that can more readily assess uncertainty in

key model components such as transport, flux fields, and background concentrations. Along with the results presented here, previous studies (e.g., Turnbull et al., 2011) provide key information that is necessary to guide and motivate more extensive future measurement and emissions evaluation efforts. Such future efforts will benefit substantially from incorporating newly available information from column-average $CO_2$ concentrations acquired by orbiting instruments or ground-based spectrometers to

increase observational coverage. A number of existing (OCO-2, OCO-3) and planned satellite missions will significantly reduce the observational gap in China, though surface observations provide additional constraints and a link to absolute calibration scales. A denser network of $CO_2$ measurement stations in China is required as a component for effective monitoring, reporting, and verification of regional and national inventories. The results of this research present a necessary baseline for a key $CO_2$-emitting

region of China. Our results have broad implications toward designing future analyses as more observations of China's $CO_2$ continue to become available, particularly in the era of increased $CO_2$ satellite coverage. However, as the quality of satellite retrievals can be compromised by factors such as aerosol loading, surface observations continue to be crucial for the region both in their own right and as a key component of cross-platform evaluations.


**Code and Data Availability**

Code and data are available through the Harvard Dataverse at https://doi.org/10.7910/DVN/OJESO0.
The code and data supplement includes observational and modeled $CO_2$ time series, WRF and STILT
parameter files, and STILT footprint files.

**Author Contributions**

A.D., J.W.M, and S.C.W designed the research. A.D. performed the research with guidance from all co-
authors. Y.W. and J.W.M monitored, maintained, and provided access to the Miyun hourly observational
data set. Y.Z. provided the China-specific anthropogenic inventory. WRF-STILT simulations were
performed by A.D. with assistance from T.N. A.D. constructed the vegetation $CO_2$ inventory. A.D. and
J.W.M wrote the paper with contributions from all other co-authors.

**Competing Interests**

The authors declare no competing interests.

**Acknowledgments**
We acknowledge the Harvard-China Project and the Harvard Global Institute for funding this study. We
thank Zhiming Kuang for providing computational resources. We also thank Jenna Samra, Maryann
Sargent, and Victoria Liublinska for helpful discussion.

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
