# Peer review of "Evaluating China's anthropogenic CO2 emissions inventories: a northern China case-study using continuous surface observations from 2005-2009."

_Atmospheric Chemistry and Physics, 2019_

## Referee Comment (RC1) · Anonymous Referee #1 · 3 Dec 2019

The article by Dayalu et al focuses on CO2 emissions and atmospheric observations for 2005-2009 in Northern China. They analyse and evaluate 3 anthropogenic inventories for the studied region, two that are subsets of global inventories (EDGAR, CDIAC) and a China specific one (ZHAO). The results highlight the importance of improving emission estimates for China in order to accurately track changes in CO2 emission reductions.

In general, the paper is scientifically sound; however, it is not what the title and the start of the Abstract promises. The authors highlight interesting plans in terms of improving/optimizing emissions over China but then focus a large portion of the writing

on defending and justifying why they weren't able to accomplish them. Based on the current text it seems like the authors are also aware that the results are a subject to a large number of limitations, and try to justify these things throughout the paper. I understand that this is work in progress and the authors were restricted by the available data resources; however, the way the authors framed the whole work, my overall take home message is that the results are just not robust enough. Overall the work should be published; however, in the current format maybe ACP is not the best choice. Having in mind that the authors stated that at this stage there are no other available observations and it is not possible to perform optimization of the emissions I see two potential pathways to go ahead with the publication of this work: 1) Re-frame the writing/aim of the work and highlight more and focus more on what you did and what information you can get from the available data (with high certainty) and not what you weren't able to do 2) Extend the work/analysis by including additional methods to evaluate and constrain regional CO2 emissions and additional inventories that potentially became available since the start of this research.

General comments:

1. Lots of focus on justifications/limitations early on in the Abstract. I suggest shortening the Abstract, more specifically the part where the authors discuss that they only have one site. Try to re-frame it so that you highlight and emphasize what you have and what you did, and not what you don't have and can't do. Moreover, the limitation of only 1 site is discussed too many times in the text, no need to repeat it over and over again.

2. I would also suggest to merge the caveats section with the conclusions. I understand that including the caveats of this study is quite important to justify the results/work/effort; however, currently the focus on the caveats puts the results in the second plan.

3. The whole text could be condensed. There are a number of places where the

authors repeat the same thing. An example: Introduction lines ~115 where describing the measurements is the same as the beginning of the CO2 observations section. No need to have all the details on both places.

4. The writing needs modification and improvement, hence the paper needs a careful reading/checking. There are a number of sentences (or parts of sentences) that are hard to follow and requires few re-reading in order for the reader to understand the message. I suggest the authors to re-write and clarify the sentences under the Specific Comments.

Specific Comments:

1. Sentences that could use some rephrasing or dividing into multiple sentences:

Line 22: Comparison of CO2 observations to CO2 predicted from accounting for global background concentration and atmospheric mixing of emissions suggests potential biases in the inventories.

Line 39: Additionally, we note that averaged over the study time period, the unscaled China-specific inventory has substantially larger annual emissions for China as a whole (20% higher) and the northern China evaluation region (30%) than the unscaled global inventories.

Line 42: lend support the rates

Line 180: Winter wheat emergence in the spring and corn emergence in later summer shift the seasonal patterns such that regional seasons are more appropriately represented when months of year are grouped as January, February, March (JFM/Winter); April, May, June (AMJ/Spring); July, August, September (JAS/Summer); and October, November, December (OND/Fall), respectively.

Line 285: This is not intended as an exhaustive sampling of inventory approaches; however, it is sufficient to demonstrate the utility of continuous high-accuracy observations as a top-down constraint for evaluating emissions estimates.

Line 468: As noted in Sect. 3, the regional growing season does not have a typical pattern in that peak uptake occurs around July/August with the onset of the corn growing season.

2. Also few wording that needs re-phrasing:

Line 45: import

Line 80: exhaustive

Line 95: judge

Line 110: while the others do not

3. More details needed:

Abstract Line 21: "CO2 inventories" – list which ones.

Line 285: "This is not intended" - this as what, the study? Clarify.

Line 320: "Applying the weekly and diurnal Nassar et al. (2013) scaling factors did not generate differences that were statistically significant, suggesting that a more rigorous set of temporal scaling factors need to be developed for China. " Is this based on work from the authors or Nassar or? Clarify.

4. Line 145: "it is not possible to evaluate any error in spatial allocation of emissions. However, we note that the same transport model is applied to all the emission fields. Unresolved transport error undoubtedly contributes to scatter in the model-data comparison but is unlikely to generate consistent biases among the inventories." - could you please explain this better.

5. Line 170: "Average annual data coverage" – was this calculated based on hourly, daily data? Just add some brief details how was it quantified.

6. Line 189: "filtered to include only non-missing observations" – a little bit unclear, does this means that only days are used when we have measurement for each hour

between 11 and 16?

7. Line 190: "background criteria" – if possible, briefly mention what it is in the main text also or additionally refer to section 3.5.

8. Methods section – when describing why the 11-16 measurements are used, please add a discussion of why the authors didn't use night time data and how much this affects the results. Although this is briefly mentioned at the end of section 3.3. it would be good to extended it in the Methods section also.

9. Figure 2: It would be good to add another sentence on what the different percentile regions represent/describe. This could be added around Line 235.

10. Line 259: "which has been noted previously as major uncertainty in Chinese emission inventories" – add reference.

Technical comments:

1. Feel free to remove the word respectively from everywhere in the text. It is already automatically assumed that the order is respectively.

2. Line 160 rephrase 'made' –> "measured"

3. Table 1. – define what the abbreviations are (if used for the first time in the text). And just to clarify, these are the 2005-2009 averages? Add in the caption.

4.Line 820: (in press), 2018 – still in press?

---

## Referee Comment (RC2) · Anonymous Referee #2 · 5 Dec 2019

Dayalu et al. 2019 compare observations of atmospheric CO2 mole fractions from the Miyun stations from 2005-2009 with a modelling framework based on the WRF-driven STILT model, biospheric CO2 fluxes from VPRM and three different spatially explicit inventories for anthropogenic CO2 emissions. The study investigates the model-observation mismatches at different temporal scales for a domain limited by the sensitivity of observations at the measurement site according to WRF-STILT. The scaled inventories are then used to assess of regional changes in carbon intensity can are visible in the 2005 to 2009 time period. The paper is overall well-written despite a sometimes complicated structure. It seems that a bigger focus should be given to the key results (annual CO2 emission increments and regional carbon intensity trends).

[Figure]

The authors address a lot of the caveats of a study that are to be expected due to the limitation of a single observational site. However, not all issues are fully addressed at this stage. Significant changes to both content and structure seem required before this manuscript could be considered for ACP.

General comments:

The title implies that actual flux estimates for Northern China are the central point of this paper. However, this paper is very technical and focuses much more on the comparison of existing inventories with atmospheric observations and also the regional emission intensity. The title should be updated to better reflect this core content.

The authors present the L_90 footprint, which usually reflects the theoretical sensitivity of the observations to a unit of flux. Two issues arise with this. First and foremost, the actual area influencing the receptor observations is not reflected by this, as CO2 sources span multiple orders of magnitude. Therefore, a major source e.g. coal-fired power plants just outside the L_90 footprint will have more influence on Miyun CO2 mole fractions than a deserted patch of land without CO2 flux inside the L_90 footprint (e.g. in inner Mongolia). The second problem is that the authors use a plethora of different terms and all seem to refer to nearly(?) the same thing: "L_90 footprint", "influence region", "L_90 region", "L_90 evaluation region", "90th percentile of multi-year mean annual STILT footprint influences", "surface influence maps".

A further limitation is that only one biosphere model is used. The authors seem to ignore this limitation, while nicely highlighting that having 3 anthropogenic prior is very helpful to better understand general results of modelled atmospheric CO2. The fact that biospheric fluxes might be even more uncertain than anthropogenic CO2 fluxes seems to unrecognized. A straightforward analysis to investigate the relevance of natural versus anthropogenic fluxes would be to investigate if the biggest model-observation mismatches systematically occurr during times of high contributions of anthropogenic or natural fluxes to the modeled CO2.

Furthermore, multiple papers that address anthropogenic CO2 emissions in China and specifically the Beijing region are ignored, e.g. the PKU-CO2 inventory (see comment line 35f) or the isotope studies by Niu et al. 2016 (see comment line 364f). A comparison to their results would be an important addition to this study. Lastly, one important result of this study is the trend in regional carbon intensity. Unfortunately, the calculation of GRP and its trend as well as their uncertainty is not clear enough. To really assess the importance of reducing the uncertainty in CO2 emission estimates by using atmospheric observations strongly depends on how well GRP and GRP trends can be calculated and also scaled to GDP and GDP trends. The explanation, data sources and methodologies for the GRP calculation should be expanded.

Specific comments:

Line 35f: When did this research begin? The PKU-CO2 emissions inventory which is China-specific was published in 2013, but is not considered or even mentioned in this manuscript (Wang et al. 2013; doi:10.5194/acp-13-5189-2013, available through e.g. http://inventory.pku.edu.cn/download/download.html)

Line 56f: The author's should expand more on the nature of the differences of different inventories. Atmospheric measurements will only be able to consider scope 1 emissions and do always include all sources, while national inventory reporting and provincial reporting might use different methodologies and also different emission category definitions and reporting thresholds. Therefore, it is unclear if the mere fact that there is a discrepancy between provincial and national estimates really means that there is a difference that an atmospheric approach could detect, help to decrease.

Line 79: see comment line 35f

Line 118: A key element that needs further explanation is the notion of "surface influence map". This seems to be used to describe the footprint, i.e. the sensitivity of the observations to a unit of flux (emission) from a given area. However, in line 645 the "L_90 footprint" is apparently something separate from the "influence region". See

general comments.

L130: Figure 10 from Dayalu et al. 2018 does indeed show that natural and anthropogenic fluxes are the same order of magnitude in the growing season. But given the very significant variability (1-sigma is near 100%) it seems unclear why the authors assume that this is only in the peak growing season and not also in other months e.g. May 2006 seems to have high uncertainties in relative importance of natural versus anthropogenic CO2 fluxes.

L146f: Please elaborate why transport errors (which can be systematic in nature) could not cause a bias when comparing inventories with distinctly different spatial distributions.

L160f: Wang et al. 2010 provide information on instrumental precision and that a calibration strategy was in place to monitor long-term drifts. This seems like an important addition here.

Line201f: Given that a short tower is used for observations it seems useful to know what the height of the lower/lowest WRF levels used are. 41 vertical levels are mentioned, but without additional information this seems difficult to interpret.

Line 234f and Figure 2: It seems important to expand on how your "L_90 region" was calculated. It is referred "90% of the surface influencing measurements". So this would mean that it is NOT the footprint or the 90th percentile of the surface sensitivity. The surface sensitivity/footprint reflects how a unit of flux will alter the observed mole fraction. However, even regions with very low sensitivity can still have a noticeable influence on the observed concentrations. Figure S8a clearly shows that some regions within the "90th percentile (Northern part of China) have emission rates that are at least 3 orders of magnitude lower than areas just South of the 90th percentile footprint (Nanjing-Shanghai region). It seems very likely that atmospheric CO2 mole fractions at Miyun would be more affected by these Southern Emissions than from some remote Northern regions that. A true influence/contribution map could be calculated very

quickly with the existing data.

Line 238f: Are really 40% influence/contribution coming from outside of L_50 or rather 40% of footprint sensitivity lies outside L_50?

Line 256f: The justification of the interpolation seems to rely on the fact that only a few regions show large differences. However, a more straightforward method would be to convolute the 2005 footprints with 2005 emissions and then with 2009 emissions (using the same 2005 footprints). This way we can directly assess if the flux changes are theoretically noticeable in the atmospheric record used later or if this just adds "random noise" to the observations.

Line 274/275: citations needed

Line 287f: see comment line 35f

Line 332f: Given that only one simple biosphere model is used in this study a discussion of its performance and uncertainty would be very useful here. How well does VPRM-China compare to the local/regional flux towers sites within the L_90 footprint?

Line 351 – eq 1? The equation seems to imply that CT2015 was used for all years – maybe add clarification that CT fluxes for the appropriate years was used and not a climatology based on CT2015. CO2(t) = CO2,obs(t)-CO2,CT2015(t-7d) Also, the equation is not numbered/labelled.

Line 364f and S5: The authors suggest that the anthropogenic fluxes dominate the annual total in the main text and then go even further in the supplement and suggest that natural fluxes are negligible. Quote from S5: ". . . correction at annual scales is therefore applied only to the anthropogenic emissions inventories" This a very strong assumption and seems to require further explanation. During the growing season fluxes seem comparable and VPRM underestimates respiration fluxes in the non-growing season (see line 504). In the absence of other biosphere models in this study to cement this notion it seems necessary to refer to other studies in China to rationalize this.

For example, Niu et al. 2016 [https://pubs.acs.org/doi/abs/10.1021/acs.est.5b02591] found that even in Beijing (Haidan district) CO2 from fossil burning only contributes 75% to the annual average CO2 offset. So, it seems unlikely the natural contribution to the CO2 mole fractions at Miyun can be ignored, even at annual average scale.

Line 373f: Why is VPRM now classified/labelled as an inventory and not as a biosphere model anymore?

Line 402f: Please clarify the distinction you make between "footprint extent" and "influence region"

Line 415 - 417: Please clarify - on one hand, during winter the receptor is predominantly influenced from low emitting regions northwest (Inner Mongolia), but also subject to CO2 from inefficient district heating? Is that district heating in Mongolia? Or do the more local CO2 sources (e.g. Beijing) dominate the atmospheric CO2 mole fractions at Miyun during this season?

Line 457f: Suggests that section 4.2.1 implies that better performance of EDGAR and CDIAC is due to an artifact of their lower emissions. However, section 4.2.1 does not explain why EDGAR+VPRM and CDIAC+VPRM being too low has to be due to EDGAR and CDIAC and not a feature of VPRM. One could also easily argue that ZHAO+VPRM matching at hourly scale is an artifact due to too high anthropogenic fluxes. A more detailed discussion why matching hourly data is correct and matching the seasonal data is likely an artifact would be helpful here. One point raised later (line 504) is that VPRM underestimates non-growing season CO2 respiration. Wouldn't this even further improve the fit of EDGAR/CDIAC+VPRM in Figure 4? And maybe partially explain the underestimation at hourly timescale?

Line 505-510: see comment line 457f.

Line 550: More detailed needed on the calculation of GRP and a proper assessment of its uncertainty is crucial, see general comments.

Line 551-556: A list of potential reasons for changes in GRP and CO2 emissions is given here, but it is unclear if this is linked to table 3 or just a list of events happening in the discussed time window. For example, the financial crisis of 2008 is mentioned, but no decrease in GRP is visible in Figure 9a.

Line 556: Should maybe refer to Figure 9a not 6a?

Line 573: Previous section was already 4.4

Line 590: Figure 8 seems not to support an evident increase in CO2 emissions strongly correlated with GRP. Maybe a scatter plot of GRP versus CO2 emissions would help to highlight a possible correlation.

Line 596: Please elaborate why doubling of GRP suggests enlarged production capacity as driver for emission reductions? Would a shift towards more service-oriented businesses or production of higher value goods have the same effect?

Line 597: The reported decrease of regional carbon intensity by 47% (28%, 65%) is based on which inventory?

Figure 9: This is a core result of the study and could be discussed in more detail. Are the regional carbon intensity trends of the 3 inventories significantly different after applying the correction when uncertainties in GRP are accounted for?

Line 608: A longer discussion of the limitations introduced by only using one biosphere model could be added.

Line 645f: see comment line 402f

---

## Author Comment (AC1) · 14 Feb 2020

We thank the two reviewers for the detailed and helpful comments which have strengthened the paper. Individual responses to each of the points raised are provided below.

Response to comments from Anonymous Referee #1 (RC1)

1. I see two potential pathways to go ahead with the publication of this work: 1) Reframe the writing/aim of the work and highlight more and focus more on what you did and what information you can get from the available data (with high certainty) and not what you weren't able to do 2) Extend the work/analysis by including additional

methods to evaluate and constrain regional CO2 emissions and additional inventories that potentially became available since the start of this research.

- We agree that the focus should be more on spotlighting the key results (pathway 1) and our revisions have focused on this. Extension to additional emissions inventories (pathway 2) without additional observations is simply throwing more modeled quantities at sparse observations (this work is observation-limited, not inventory-limited). In this context, a selection of three inventories is sufficient to make our main points which are (i) the power of top-down constraints on emissions assessments in the region, motivating new ground-based observation sites; (ii) there are substantial differences in categories of inventories for China, and these broad categories (China-specific ones that are tuned to actual but difficult to obtain field data, or global subsets based on coarser but more readily available inputs) produce significantly different observational mismatch, highlighting the importance of China-specific field data; (iii) our work suggests that current emissions tracking at the international level based on archaic inventories such as CDIAC can be very different from emissions reality; more ground-based observations are needed (along with remotely sensed observations, which now have a robust timeseries) to test this hypothesis. When sufficiently dense observations are available, a truly comprehensive analysis of all available and relevant inventories can be conducted. Such an analysis is different from the main point of our study.

2. Lots of focus on justifications/limitations early on in the Abstract. I suggest shortening the Abstract, more specifically the part where the authors discuss that they only have one site. Try to re-frame it so that you highlight and emphasize what you have and what you did, and not what you don't have and can't do. Moreover, the limitation of only 1 site is discussed too many times in the text, no need to repeat it over and over again.

- Agreed. Combining this with (1) above, we have gone through the text and deleted redundant caveats and spotlighted areas of greater certainty so they are not lost in the "caveat weeds". We also recognize there is a certain amount of subjectivity sur-

rounding how much is too much for caveats, as one of the main concerns of previous reviewers was that there were not enough.

3. I would also suggest to merge the caveats section with the conclusions. I understand that including the caveats of this study is quite important to justify the results/work/effort; however, currently the focus on the caveats puts the results in the second plan.

- Agreed. The stand-alone caveats have now been ingested into the Conclusions. Along with (1) and (2) above, we have pared down the amount of in-line caveats as well.

4. The whole text could be condensed. There are a number of places where the authors repeat the same thing. An example: Introduction lines âĹij115 where describing the measurements is the same as the beginning of the CO2 observations section. No need to have all the details on both places.

- Agreed. We have fixed this.

5. The writing needs modification and improvement, hence the paper needs a careful reading/checking. There are a number of sentences (or parts of sentences) that are hard to follow and requires few re-reading in order for the reader to understand the message. I suggest the authors to re-write and clarify the sentences under the Specific Comments.

- See responses to Specific Comments.

6. Rephrasing/restructuring. Line 22: Comparison of CO2 observations to CO2 predicted from accounting for global background concentration and atmospheric mixing of emissions suggests potential biases in the inventories

- Fixed. We have made significant changes to the abstract wording to both spotlight our key results and also improve the structure.

7. Rephrasing/restructuring. Line 39: Additionally, we note that averaged over the study time period, the unscaled China-specific inventory has substantially larger annual emissions for China as a whole (20% higher) and the northern China evaluation region (30%) than the unscaled global inventories.

- Fixed. We have made significant changes to the abstract wording to both spotlight our key results and also improve the structure.

8. Rephrasing/restructuring. Line 42: lend support the rates

- Fixed. We have made significant changes to the abstract wording to both spotlight our key results and also improve the structure.

9. Rephrasing/restructuring. Line 180: Winter wheat emergence in the spring and corn emergence in later summer shift the seasonal patterns such that regional seasons are more appropriately represented when months of year are grouped as January, February, March (JFM/Winter); April, May, June (AMJ/Spring); July, August, September (JAS/Summer); and October, November, December (OND/Fall), respectively.

- Fixed.

10. Rephrasing/restructuring. Line 285: This is not intended as an exhaustive sampling of inventory approaches; however, it is sufficient to demonstrate the utility of continuous high-accuracy observations as a top-down constraint for evaluating emissions estimates.

- Fixed.

11. Line 468: As noted in Sect. 3, the regional growing season does not have a typical pattern in that peak uptake occurs around July/August with the onset of the corn growing season.

- Fixed.

12. Rephrasing. Line 45: import

- Fixed.

13. Rephrasing. Line 80: exhaustive

- We use the word exhaustive in response to previous reviewers who were concerned that we were claiming the three inventories were the only ones that existed for China. We wanted to assure readers that we recognize that these are the only inventories that exist for China. We feel this word should be left in for that reason.

14. Rephrasing. Line 95: judge

- We use this word in response to previous reviewers who felt we were actually judging the merits of the inventories individually. That has not been our intent. We feel this word should be left in for that reason.

15. Rephrasing. Line 110: while the others do not

- Fixed.

16. More details needed. Abstract Line 21: "CO2 inventories" – list which ones.

- We are leaving the named inventories out of the abstract in response to previous reviewers who felt that calling them out specifically at that stage implied criticizing the particular inventories rather than a generalized examination of the approaches they represented. We feel the wording should remain this way until the reader has the deeper context from the text.

17. More details needed. Line 285: "This is not intended" - this as what, the study? Clarify.

- Fixed – "Our study is not".

18. More details needed. Line 320: "Applying the weekly and diurnal Nassar et al. (2013) scaling factors did not generate differences that were statistically significant, suggesting that a more rigorous set of temporal scaling factors need to be developed

for China. " Is this based on work from the authors or Nassar or? Clarify.

- It was based on our work. We have clarified this.

19. Line 145: "it is not possible to evaluate any error in spatial allocation of emissions. However, we note that the same transport model is applied to all the emission fields. Unresolved transport error undoubtedly contributes to scatter in the model-data comparison but is unlikely to generate consistent biases among the inventories." - could you please explain this better.

- We have reworded and expanded this section. We have also restructured the paragraph to make the point clearer.

20. Line 170: "Average annual data coverage" – was this calculated based on hourly, daily data? Just add some brief details how was it quantified.

- Fixed. (Calculated based on hourly data).

21. Line 189: "filtered to include only non-missing observations" – a little bit unclear, does this means that only days are used when we have measurement for each hour between 11 and 16?

- We have clarified this. The subset is done for each individual hour, not the daily blocks. For example, if we were missing 1100h but had 1200-1600 for a particular day, 1200-1600 would be used but 1100 would not (because we would not be able to compare our modeled quantities for that hour to any observation). But we would be able to compare modeled to obs for 1200-1600h on that same day.

22. Line 190: "background criteria" – if possible, briefly mention what it is in the main text also or additionally refer to section 3.5.

- Fixed. Referring to section 3.5, where we also included a very brief discussion of what that criteria is (previously mentioned only in the SI).

23. Methods section – when describing why the 11-16 measurements are used, please

add a discussion of why the authors didn't use night time data and how much this affects the results. Although this is briefly mentioned at the end of section 3.3. it would be good to extended it in the Methods section also.

- We do explain this in that paragraph (∼L190) but we have improved the wording to make it more clear as to why we were not using nighttime data.

24. Figure 2: It would be good to add another sentence on what the different percentile regions represent/describe. This could be added around Line 235.

- We have considerably re-worded our presentation of percentile regions in response to concerns from the second reviewer.

25. Line 259: "which has been noted previously as major uncertainty in Chinese emission inventories" – add reference.

- Reference added.

26. Feel free to remove the word respectively from everywhere in the text. It is already automatically assumed that the order is respectively.

- The comment is appreciated, but I would feel more comfortable keeping "respectively" in formal writing as is customary when multiple variables are being described. Even if the order is understood by most people, it eliminates confusion more than it distracts.

27. Line 160 rephrase 'made' –> "measured'

- Fixed.

28. Table 1. – define what the abbreviations are (if used for the first time in the text). And just to clarify, these are the 2005-2009 averages? Add in the caption.

- Fixed.

29. Line 820: (in press), 2018 – still in press?

- No longer in press. Fixed.
Response to comments from Anonymous Referee #2 (RC2)

1. The title implies that actual flux estimates for Northern China are the central point of this paper. However, this paper is very technical and focuses much more on the comparison of existing inventories with atmospheric observations and also the regional emission intensity. The title should be updated to better reflect this core content.

- Agreed. We have changed the title accordingly. It is now "Evaluating China's anthropogenic $CO_2$ emissions inventories: a northern China case-study using continuous surface observations from 2005-2009."

2. The authors present the L_90 footprint, which usually reflects the theoretical sensitivity of the observations to a unit of flux. Two issues arise with this. First and foremost, the actual area influencing the receptor observations is not reflected by this, as $CO_2$ sources span multiple orders of magnitude. Therefore, a major source e.g. coal-fired power plants just outside the L_90 footprint will have more influence on Miyun $CO_2$ mole fractions than a deserted patch of land without $CO_2$ flux inside the L_90 footprint (e.g. in inner Mongolia). The second problem is that the authors use a plethora of different terms and all seem to refer to nearly(?) the same thing: "L_90 footprint", "influence region", "L_90 region", "L_90 evaluation region", "90th percentile of multiyear mean annual STILT footprint influences", "surface influence maps".

- We recognize there has been a misunderstanding about the role of the L_0.90 region in our study. We apologize for the confusion caused by the way this concept was explained and introduced, especially with the effect of inconsistent references (we now consistently use L_0.90 region). We have explained the role of the L_0.90 region and the methodology for calculation more thoroughly in the text. In particular, we have clearly stated that at any given time the *entire* STILT footprint is convolved with the flux estimates; the L_0.90 region is simply the region we chose to ascribe the model-observation mismatch. The area of the region informs the conversion of ppm mismatch to mass units and we compare this regionally scaled mass correction to the mass

originally estimated by the fluxes in that L_0.90 bounding area. We could have just as easily made our bounding area for evaluation of mismatch be L_0.75 or L_0.99 but we explain in the text why the L_0.90 region is a good balance of capturing enough of the surface sensitivity with not having an unrealistically diffuse spatial area. With this clarification in mind, your concern about neglecting influential sources is alleviated: we *are* taking such sources into account in the main footprint*flux = ppm setup; we just ascribe that ppm to the L_0.90 region in the end to obtain a mass correction over a reasonably influential area. The data set does not really allow us to geographically allocate the mismatch beyond this relatively coarse method.

3. A further limitation is that only one biosphere model is used. The authors seem to ignore this limitation, while nicely highlighting that having 3 anthropogenic prior is very helpful to better understand general results of modelled atmospheric CO2. The fact that biospheric fluxes might be even more uncertain than anthropogenic CO2 fluxes seems to unrecognized. A straightforward analysis to investigate the relevance of natural versus anthropogenic fluxes would be to investigate if the biggest model-observation mismatches systematically occurr during times of high contributions of anthropogenic or natural fluxes to the modeled CO2.

- We have modified the text accordingly. We used only one biosphere model to simplify our assessment of the variations across different anthropogenic emissions inventories (and we have changed the paper title accordingly). Our companion paper (Dayalu et al., 2018) highlights differences across vegetation models when controlling for anthropogenic emissions. We did not intend to imply that only the anthropogenic inventories are needed to understand China's atmospheric CO2 and we have reworded the text to make this clearer. In addition, we make more references to the Dayalu et al. (2018) VPRM-CHINA paper (in agreement with the Turnbull et al. (2011) paper) that specifically highlights that in the heavily agricultural region of the North China Plain, the *peak* growing season sink actually is comparable in magnitude to the anthropogenic source. Outside of this observation, we do state that the Summer-time analysis of an-

thropogenic vs biogenic cannot really be undertaken absent additional and diverse data sets. Table 1 (Mean Bias and RMSE segment) does highlight the model-observation mismatch by season, as you had suggested. In the winter in northern China, the anthropogenic signal is the dominant signal, swamping the NEE terms. The systematic bias among the three simulations in this season (Table 1) is largely attributable to differences in the anthropogenic emissions alone (all other modeled components being identical among the three modeled $CO_2$ quantities). Based on the results presented in Table 1 (with the confounding exception of the Summer for reasons we just described) we see that during seasons where biological activity is lower or significantly lower than anthropogenic activity, there is a consistent discrepancy between the $CO_2$ modeled by the three different anthropogenic inventories suggesting a systematic difference in the anthropogenic component. In the fall, all three modeled quantities are consistently lower than observations most likely resulting from the known underestimate of ecosystem respiration which is the dominant biological process at this season (Dayalu et al., 2018); but even so China's significant anthropogenic component still dominates at this time. If we assume that the winter represents the "purely anthropogenic" baseline, and we assume a certain percentage impact of temporal activity factors (1.5-8ppm as suggested by Nassar et al. 2013) we could make an estimate as to how much this baseline is expected to shift over the course of seasons – but that would be overextending our analysis as we know very little about temporal activity factors in China.

4. Furthermore, multiple papers that address anthropogenic $CO_2$ emissions in China and specifically the Beijing region are ignored, e.g. the PKU-CO2 inventory (see comment line 35f) or the isotope studies by Niu et al. 2016 (see comment line 364f). A comparison to their results would be an important addition to this study. Lastly, one important result of this study is the trend in regional carbon intensity. Unfortunately, the calculation of GRP and its trend as well as their uncertainty is not clear enough. To really assess the importance of reducing the uncertainty in $CO_2$ emission estimates by using atmospheric observations strongly depends on how well GRP and GRP trends can be calculated and also scaled to GDP and GDP trends. The explanation, data

sources and methodologies for the GRP calculation should be expanded.

- Re: inventories. See responses to 5 and 20 below. Re: GRP calculations: we have expanded the text and descriptions of the calculations.

5. Line 35f: When did this research begin? The PKU-CO2 emissions inventory which is China-specific was published in 2013, but is not considered or even mentioned in this manuscript (Wang et al. 2013; doi:10.5194/acp-13-5189-2013, available through e.g. http://inventory.pku.edu.cn/download/download.html)

- We include this paper in the references now (and cite it as an additional justification for our not explicitly separating ODIAC and CDIAC for China – a major concern of previous reviewers). That aside, the PKU-CO2 emissions inventory referred to in the Wang et al., 2013 paper was a *global* emissions inventory solely for the year 2007. We selected the Zhao inventory due to the fact that at the time the study was conducted, it was the only readily available China-specific inventory that spanned our observational data set (2005-2009). Furthermore, the paper does specify use of the CARMA power plant inventory for emissions factors; for reasons described in the text, the global emissions factors provided by CARMA for China are known to be problematic. PKU and MEIC have since been leaders in developing China-specific inventories, but unfortunately the study concluded before those were readily available and ingestible into our analysis framework. We agree that future studies in China would benefit greatly from more China-specific inventories being evaluated, and look forward to results from such analysis when longer timeseries of observational data become available.

6. Line 56f: The author's should expand more on the nature of the differences of different inventories. Atmospheric measurements will only be able to consider scope 1 emissions and do always include all sources, while national inventory reporting and provincial reporting might use different methodologies and also different emission category definitions and reporting thresholds. Therefore, it is unclear if the mere fact that there is a discrepancy between provincial and national estimates really means that

there is a difference that an atmospheric approach could detect, help to decrease.

- Agreed – the observations give an overall constraint on total fluxes and cannot re-solve the particulars of estimate methodology. We are looking at whether totals are consistent with observations, but we can't really diagnose the finer scale methodology (we can't attribute the error to any particular source). That being said, if there is a significant difference between the totals reported by provincial vs national inventory estimates then the discrepancy can be suggestive of differences in methodology. With our approach we can assess whether total fluxes over a region are consistent with ob-servations, but with only one species we can't really diagnose which emission source types are too high or too low. One inventory is closest to matching the observations than the other, but we can't say what feature makes this so. The most we can do is highlight the major differences among the inventory methodologies (as we have done in Section 3.3); we don't know which of these changes accounts for the better model-observation match. We have made it clearer in the introduction to ensure we are not suggesting this. "The primary intent of the comparisons presented here is not to judge specific inventories, but to demonstrate that even a single site with a long record of high time resolution observations can identify the potential impact of major differences among inventories that manifest as biases in the model-data comparison."

7. Line 79: see comment line 35f

- See response #5.

8. Line 118: A key element that needs further explaination is the notion of "surface influence map". This seems to be used to describe the footprint, i.e. the sensitivity of the observations to a unit of flux (emission) from a given area. However, in line 645 the "L_90 footprint" is apparently something separate from the "influence region". See general comments.

- Yes – we have fixed this in the text and included a new set of figures in the SI to further explain the footprint map/notion of surface influences.

9. L130: Figure 10 from Dayalu et al. 2018 does indeed show that natural and an-thropogenic fluxes are the same order of magnitude in the growing season. But given the very significant variability (1-sigma is near 100%) it seems unclear why the authors assume that this is only in the peak growing season and not also in other months e.g. May 2006 seems to have high uncertainties in relative importance of natural versus anthropogenic $CO_2$ fluxes.

- Uncertainty in the modeled biosphere is undoubtedly significant, but contributes equally to the variability across all the modeled-observation quantities. We have made this clearer in the uncertainty discussion as well. See response to #20.

10. L146f: Please elaborate why transport errors (which can be systematic in nature) could not cause a bias when comparing inventories with distinctly different spatial distributions.

- We have amended the line to include that it may be important in that it could attribute errors in transport to biases in inventories. Although the interaction of transport error with differences in spatial distribution could bias individual observations, averaging over longer timescales (seasons, years) minimizes the bias of individual points. We have made this clearer in the text.

11. L160f: Wang et al. 2010 provide information on instrumental precision and that a calibration strategy was in place to monitor long-term drifts. This seems like an important addition here.

- Fixed. We also note the citation for Wang et al., 2010 at the end of the section directing the readers to that paper for details on the instrument precision, calibration strategy, etc.

12. Line201f: Given that a short tower is used for observations it seems useful to know what the height of the lower/lowest WRF levels used are. 41 vertical levels are mentioned, but without additional information this seems difficult to interpret.

- We use the default WRF eta levels generated with the 41-level specification and we would expect about 20 vertical levels in the first 1500m. Our first vertical level would be roughly around 8m (using Arasa et al., 2016 as a guide: https://www.scirp.org/pdf/ACS_2016042911473822.pdf). We also note this is the other reason for restricting our analysis to middle of day – excluding times when the vertical gradients are sharp.

13. Line 234f and Figure 2: It seems important to expand on how your "L_90 region" was calculated. It is referred "90% of the surface influencing measurements". So this would mean that it is NOT the footprint or the 90th percentile of the surface sensitivity. The surface sensitivity/footprint reflects how a unit of flux will alter the observed mole fraction. However, even regions with very low sensitivity can still have a noticeable influence on the observed concentrations. Figure S8a clearly shows that some regions within the "90th percentile (Northern part of China) have emission rates that are at least 3 orders of magnitude lower than areas just South of the 90th percentile footprint (Nanjing-Shanghai region). It seems very likely that atmospheric $CO_2$ mole fractions at Miyun would be more affected by these Southern Emissions than from some remote Northern regions that. A true influence/contribution map could be calculated very quickly with the existing data.

- Agreed – the concept was poorly explained which has led to the misunderstanding. We have now explained our methodology better and included footprint maps and percentile selection illustration in the SI. See detailed comments to your previous point (#2.)

14. Line 238f: Are really 40% influence/contribution coming from outside of L_50 or rather 40% of footprint sensitivity lies outside L_50?

- Agreed – again, poorly worded. This has been fixed. We also modified to state L_0.75, as this seemed more interesting a comparison.

15. Line 256f: The justification of the interpolation seems to rely on the fact that only

a few regions show large differences. However, a more straightforward method would be to convolute the 2005 footprints with 2005 emissions and then with 2009 emissions (using the same 2005 footprints). This way we can directly assess if the flux changes are theoretically noticeable in the atmospheric record used later or if this just adds "random noise" to the observations.

- While the test proposed would be interesting, evaluating errors in spatial allocation is beyond the scope of this study and the available data set (L143). It would still be relying on a single site to optimize a spatial distribution of emissions and would not be a conclusive test; our approach was the simplest method using the information provided by the raw anthropogenic inventory alone.

16. Line 274/275: citations needed

- Fixed.

17. Line 287f: see comment line 35f

- See response 5.

18. Line 332f: Given that only one simple biosphere model is used in this study a discussion of its performance and uncertainty would be very useful here. How well does VPRM-China compare to the local/regional flux towers sites within the L_90 footprint?

- We direct readers to the companion paper by Dayalu et al (2018) which shows these figures (E.g., Figure 5). Eddy flux sites in the region are sparse, and most of the available data was enough to be used only as calibration. Only two sites had enough data to be used for validation.

19. Line 351 – eq 1? The equation seems to imply that CT2015 was used for all years – maybe add clarification that CT fluxes for the appropriate years was used and not a climatology based on CT2015. CO2(t) = CO2,obs(t)-CO2,CT2015(t-7d) Also, the equation is not numbered/labelled.

- The equation is now labeled. CT2015 is the CarbonTracker version number (not the year of the data). CT2015 was used for all years (ie. that version of CarbonTracker), and we used atmospheric mixing ratios not fluxes. Setting (t-7d) in the subscript implies that was always used – however, as we detail in the supplementary information (and now in the text) background concentrations were selected when the particle reached the domain edges which may or may not be as far back as 7days (7 days being the backward limit).

20. Line 364f and S5: The authors suggest that the anthropogenic fluxes dominate the annual total in the main text and then go even further in the supplement and suggest that natural fluxes are negligible. Quote from S5: ". . . correction at annual scales is therefore applied only to the anthropogenic emissions inventories" This a very strong assumption and seems to require further explanation. During the growing season fluxes seem comparable and VPRM underestimates respiration fluxes in the non-growing season (see line 504). In the absence of other biosphere models in this study to cement this notion it seems necessary to refer to other studies in China to rationalize this. For example, Niu et al. 2016 [https://pubs.acs.org/doi/abs/10.1021/acs.est.5b02591] found that even in Beijing (Haidan district) CO2 from fossil burning only contributes 75% to the annual average CO2 offset. So, it seems unlikely the natural contribution to the CO2 mole fractions at Miyun can be ignored, even at annual average scale.

- The data from the two sites in Niu et al. study is only for one year (2014), and the contribution of fossil fuels to the Beijing site displays considerable variance (75% +/- 15%); nevertheless we have incorporated these important results in our paper to caveat our statements about annual emissions but we also note that the timing relative to our study (2014 vs 2005-2009) and the variance (60% - 90% contributed by fossil fuels) annually. In line 364 we don't say the biospheric impact if zero; we say the anthropogenic signal dominates (which is true, also according to Niu et al's results). More biospheric models are needed to quantify the regional biospheric impacts and

we note this in the conclusions. This is an area of significant uncertainty, as evidenced in Piao et al. (2009) which we also cite at line 478 ("For annual budgeting we follow the assumptions of Piao et al. (2009) and Jiang et al. (2016) that agricultural systems are in annual carbon balance because crop biomass has a short residence time.") The quantity relevant to this question is the annual *net* biospheric carbon flux: and annual net carbon balance in this region is highly uncertain with an uncertainty in both magnitude and sign (ie, spanning zero) both for process-based models and inversions (Piao et al. 2009). Piao et al examine this quantity by region of China, noting that the uncertainty is very large (his regional inversions are based on 9 sites across all of Asia). Process-based models and prior models corresponding to our northern China study region (Figure 2, Piao et al) assume either small net emissions or zero (ie. zero in the agriculturally dominated north china plain). To the extent that there is a net biosphere source/sink at the annual scale, it should be included but is currently highly uncertain. Our assumption of dominant anthropogenic influence in northern china is in keeping with the priors (e.g. from Piao et al.) that assume zero and are not significantly corrected by the poorly constrained inversions. We have summarized this discussion in the text.

21. Line 373f: Why is VPRM now classified/labelled as an inventory and not as a biosphere model anymore?

- We have changed the wording to make it clearer (the VPRM model output is biogenic fluxes of CO2).

22. Line 402f: Please clarify the distinction you make between "footprint extent" and "influence region"

- We have reworded substantially to have consistent phrasing, and have made the linkage clear where we interchange surface influence and footprint. Furthermore, we are now consistent with how we refer to the L_0.90 region without mistakenly overstating its role in our study.

23. Line 415 - 417: Please clarify - on one hand, during winter the receptor is predominantly influenced from low emitting regions northwest (Inner Mongolia), but also subject to $CO_2$ from inefficient district heating? Is that district heating in Mongolia? Or do the more local $CO_2$ sources (e.g. Beijing) dominate the atmospheric $CO_2$ mole fractions at Miyun during this season?

- Our wording was confusing. As in all seasons the closer the sources are, the more influence they have. We reword to remove dominant, and instead discuss their influence on the site relative to their influence at other times of the year.

24. Line 457f: Suggests that section 4.2.1 implies that better performance of EDGAR and CDIAC is due to an artifact of their lower emissions. However, section 4.2.1 does not explain why EDGAR+VPRM and CDIAC+VPRM being too low has to be due to EDGAR and CDIAC and not a feature of VPRM. One could also easily argue that ZHAO+VPRM matching at hourly scale is an artifact due to too high anthropogenic fluxes. A more detailed discussion why matching hourly data is correct and matching the seasonal data is likely an artifact would be helpful here. One point raised later (line 504) is that VPRM underestimates non-growing season $CO_2$ respiration. Wouldn't this even further improve the fit of EDGAR/CDIAC+VPRM in Figure 4? And maybe partially explain the underestimation at hourly timescale?

- If we (justifiably) view wintertime as the anthropogenic baseline where neither GPP nor R are contributing appreciably to the signal, we see there is a consistent offset in the bias relative to observations (mean bias: ZHAO=0.01, EDGAR=-2.2, CDIAC=-3.1). We have modified the text accordingly to better illustrate the discussion that anthropogenic discrepancies are contributing to the model-observation mismatch. In any case, with the limited data and the lack of temporal activity factors we agree with your point of the ZHAO+VPRM providing too high anthropogenic emissions. We have incorporated this into the text. We have also clarified the statement at L504: VPRM underestimates respiration across the board (barring winter) not just non-growing season respiration. It's just that the effects of this underestimated respiration are more pronounced at the time of year where R is high, with lower GPP (e.g, Fall). We also wish to clarify that we are certainly not saying "hourly data is best"...we are just identifying the model the model that minimizes errors at all timescales (a multiple constraint).

25. Line 505-510: see comment line 457f.

- See response 24. We have also expanded this part accordingly.

26. Line 550: More detailed needed on the calculation of GRP and a proper assessment of its uncertainty is crucial, see general comments.

- The GRP for each province was retrieved from the IMF, World Bank, and China Statistical Yearbook. We added the GRP for each province contained in the L_0.90 region. The GRP/GPP values are very uncertain quantities, but in the form they are broadly disseminated it is unfortunately a single value for each province per year. An economic analysis to estimate uncertainty of these values is beyond the scope of our expertise and the study itself. We have, however, made our methodology clearer.

27. Line 551-556: A list of potential reasons for changes in GRP and CO2 emissions is given here, but it is unclear if this is linked to table 3 or just a list of events happening in the discussed time window.For example, the financial crisis of 2008 is mentioned, but no decrease in GRP is visible in Figure 9a.

- We plot the percent contribution to total GDP for this reason (the raw numbers alone don't necessarily provide the whole picture). In 2008 we do see a plateauing of the regional percent contribution to China's total GDP. We have made this section clearer in terms of strength of conclusions that can be drawn. We also moved the location of the sentence to make the transition from Table 3 and CO2 growth rate discussions to GRP clear.

28. Line 556: Should maybe refer to Figure 9a not 6a?

- Yes—should be 9a. Fixed.

29. Line 573: Previous section was already 4.4

- Fixed

30. Line 590: Figure 8 seems not to support an evident increase in CO2 emissions strongly correlated with GRP. Maybe a scatter plot of GRP versus CO2 emissions would help to highlight a possible correlation.

- We have included a scatterplot in Figure 9.

31. Line 596: Please elaborate why doubling of GRP suggests enlarged production capacity as driver for emission reductions? Would a shift towards more service-oriented businesses or production of higher value goods have the same effect?

- We have expanded this section and incorporated your point that the same effect would be had with service-oriented shifts.

32. Line 597: The reported decrease of regional carbon intensity by 47% (28%, 65%) is based on which inventory?

- In the text in the same sentence it states it is calculated by pooling the values across the scaled inventories: "From 2005 to 2009, carbon intensity for the L_0.90 region decreased by 47% (28%,65%), based on a one-sample t-test of pooled emissions intensity changes across scaled inventories."

33. Figure 9: This is a core result of the study and could be discussed in more detail. Are the regional carbon intensity trends of the 3 inventories significantly different after applying the correction when uncertainties in GRP are accounted for?

- See response #26: accounting for uncertainties in GRP requires access to a level of economic data that is not readily available. We have highlighted the inherent uncertainty in our economic evaluation. We have also expanded the discussion and conclusions from Figure 9 to highlight the truly interesting and main point from this which is that carbon intensity reductions and absolute carbon emissions reductions particularly

in emerging countries can be at odds with each other, and therefore distracting from climate goals.

34. Line 608: A longer discussion of the limitations introduced by only using one biosphere model could be added.

- We have incorporated more of a discussion in the uncertainty section of the conclusions.

35. Line 645f: see comment line 402f

- Fixed. See response to #2.

―――――――――――――――――――――

---

## Author Comment (AC2) · 18 Feb 2020

\*\*\*\* ADDENDUM TO AUTHOR'S COMMENTS \*\*\*\*\*

Edited response to Reviewer #2 Question #12:

Line201f: Given that a short tower is used for observations it seems useful to know what the height of the lower/lowest WRF levels used are. 41 vertical levels are mentioned, but without additional information this seems difficult to interpret.

- The first few layers of the 41 wrf-modeled vertical eta levels used by the STILT vertical coordinate system (converted to meters AGL) are provided below, noting that the miyun

receptor is within the first layer at 6magl (158masl). There are 9 vertical levels in the first 1500m.

eta mAGL

0.996500015 27

0.987999976 92

0.976500034 180

0.962000012 293

0.944000006 435

0.921499968 615

0.894500017 836

0.866361856 1070

0.839085698 1302

0.811809599 1540